# Olfactory receptor accessory proteins play crucial roles in receptor function and gene choice

Ruchira Sharma[1], Yoshiro Ishimaru[1,2], Ian Davison[3], Kentaro Ikegami[1,4], Ming-Shan Chien[1], Helena You[1], Quiyi Chi[1], Momoka Kubota[1], Masafumi Yohda[4], Michael Ehlers[5,6], Hiroaki Matsunami[1,5,7]*

[1]Department of Molecular Genetics and Microbiology, Duke University Medical Center, Durham, United States; [2]Department of Applied Biological Chemistry, Graduate School of Agricultural and Life Sciences, The University of Tokyo, Tokyo, Japan; [3]Department of Biology, Boston University, Boston, United States; [4]Tokyo University of Agriculture and Technology, Tokyo, Japan; [5]Department of Neurobiology, Duke University Medical Center, Durham, United States; [6]Biogen Inc, Cambridge, United States; [7]Duke Institute for Brain Sciences, Durham, United States

*For correspondence: hiroaki. matsunami@duke.edu

**Competing interests:** The authors declare that no competing interests exist.

**Abstract** Each of the olfactory sensory neurons (OSNs) chooses to express a single G protein-coupled olfactory receptor (OR) from a pool of hundreds. Here, we show the receptor transporting protein (RTP) family members play a dual role in both normal OR trafficking and determining OR gene choice probabilities. *Rtp1* and *Rtp2* double knockout mice (RTP1,2DKO) show OR trafficking defects and decreased OSN activation. Surprisingly, we discovered a small subset of the ORs are expressed in larger numbers of OSNs despite the presence of fewer total OSNs in RTP1,2DKO. Unlike typical ORs, some overrepresented ORs show robust cell surface expression in heterologous cells without the co-expression of RTPs. We present a model in which developing OSNs exhibit unstable OR expression until they choose to express an OR that exits the ER or undergo cell death. Our study sheds light on the new link between OR protein trafficking and OR transcriptional regulation.

## Introduction

Seven transmembrane G-protein coupled receptors (GPCRs), are diverse and the largest superfamily of receptors. Their roles are well established in sensing various stimuli including odorants, tastants, light, hormones, neurotransmitters and proteins. Some GPCRs require the presence of specific accessory proteins such as chaperones, vesicular targeting molecules and co-receptors for their cell surface expression (*Lu et al., 2003*; *Salahpour et al., 2004*; *Dey and Matsunami, 2011*; *Wu et al., 2003*). Mammalian olfactory receptors (ORs), which are GPCRs (*Buck and Axel, 1991*), are retained in the ER when expressed in non-olfactory cells. RTP1 (Receptor Transporting Protein 1) and RTP2 (Receptor Transporting Protein 2), both single transmembrane proteins strongly and exclusively expressed in the peripheral olfactory organs (*Lu et al., 2003*; *Saito et al., 2004*; *Zhuang and Matsunami, 2008*; *Gimelbrant et al., 1999*), greatly enhance the trafficking of ORs to the cell surface of heterologous cells. However, the role played by the RTPs in vivo remains unclear.

The mouse genome encodes over one thousand intact OR genes (*Niimura et al., 2014*), which are expressed in a singular and monoallelic fashion in each olfactory sensory neuron (OSN) (*Shykind et al., 2004*; *Chess et al., 1994*; *Serizawa et al., 2000*; *Malnic et al., 1999*). Each OR is

**eLife digest** Olfaction, or the sense of smell, is perhaps the most complicated and least understood of the five basic senses. Olfactory neurons in the nose can detect and distinguish between tens of thousands of different odor producing substances. They do so by using hundreds of unique sensors called olfactory receptors, each of which responds to a specific type of odor. During development, each olfactory neuron "chooses" to produce only one type of olfactory receptor. Once the neuron recognizes that the functional receptor is being generated and transported to the cell surface, it will stop making all the other olfactory receptors.

Chaperone proteins are responsible for transporting many olfactory receptors to the cell surface. To investigate how the loss of these chaperones affects how the olfactory system develops, Sharma et al. studied mice that were unable to produce the olfactory chaperone proteins. In these mice, developing neurons that chose to produce a type of olfactory receptor that depends on chaperone protein transport could not fully shut off other olfactory receptor genes. This led either to the neuron attempting to produce another type of receptor, or the death of the neuron. As a result, more neurons than usual produced receptors that do not require chaperone proteins to transport them to the cell surface. The olfactory neurons therefore produced only a fraction of all possible olfactory receptors, which decreased the ability of the mice to respond to odors.

In the future, it will be important to understand what determines whether an olfactory receptor can be transported to the cell membrane in the absence of chaperone proteins. Olfactory receptors are G protein-coupled receptors (GPCRs), which are the largest molecular class of drug targets for cancer and diseases that affect the brain and heart. Thus, results presented by Sharma et al. will also be relevant to researchers who study how GPCR malfunction causes diseases.

not chosen at random; rather, OSNs express different ORs with dramatically varying probabilities (*Khan et al., 2011*; *Ibarra-Soria et al., 2014*).

OSNs in the olfactory epithelium (OE) are organized in overlapping zones defined by the expression of each OR (*Ibarra-Soria et al., 2014*; *Ressler et al., 1993*; *Vassar et al., 1993*; *Miyamichi et al., 2005*; *Kanageswaran et al., 2015*; *Saraiva et al., 2015*) as well as in a pseudostratified manner with progenitor cells forming the basal layer and mature neurons forming the upper layers. Mature OSN dendrites project into the nasal cavity forming a dendritic knob at the surface of the OE where they express ORs to interact with odorant molecules. Mature OSN axons expressing the same OR project to the olfactory bulb (OB) to converge onto specific glomeruli (*Mombaerts et al., 1996*; *Ressler et al., 1994*; *Vassar et al., 1994*; *Hayar et al., 2004*; *Aungst et al., 2003*; *Gire et al., 2012*). When the $\beta_2$ Adrenergic Receptor ($\beta_2$AR) is expressed instead of an OR, the $\beta_2$AR –expressing OSNs target their axons to the OB and form glomeruli (*Feinstein et al., 2004a*, *2004b*; *Omura et al., 2014*; *Nakashima et al., 2013*). Hence, the development of the peripheral olfactory system is dependent on functional GPCRs.

The mechanisms by which an OSN makes an OR choice have not been fully elucidated. Locus control-region like enhancers scattered on the genome and relative location of ORs from these elements have important roles in determining the probabilities of OR gene choice (*Khan et al., 2011*; *Serizawa et al., 2003*; *Markenscoff-Papadimitriou et al., 2014*). Various epigenetic mechanisms, for example histone modification by lysine demethylase (LSD1), allow the escape of an OR gene from repression (*Magklara et al., 2011*; *Lyons et al., 2013*, *2014*; *Armelin-Correa et al., 2014*; *Kilinc et al., 2016*). OR expression is unstable until one OR is functionally expressed which then represses the expression of other OR alleles via negative feedback signaling through the unfolded protein response (UPR) and G proteins (*Serizawa et al., 2003*; *Dalton et al., 2013*; *Wang et al., 2012*; *Li and Matsunami, 2013*; *Lewcock and Reed, 2004*; *Ferreira et al., 2014*; *Nguyen et al., 2007*; *Abdus-Saboor et al., 2016*).

Here, we generated *Rtp1* and *Rtp2* double knockout mice (RTP1,2DKO) to investigate their role in the functioning and development of the olfactory system in vivo. We show that the RTP1,2DKO have OR trafficking defects, a substantial reduction in the number of mature OSNs, and an overall diminished olfactory capacity. Unexpectedly, we found that some ORs are overrepresented (referred

to as oORs) while others are underrepresented (referred to as uORs) in RTP1,2DKO. Cells expressing a uOR lack stable gene choice in the mutant compared to wild-types while cells expressing an oOR do not show this instability, a result that links OR protein trafficking and OR transcriptional regulation.

## Results

### Generation of RTP1,2DKO mice

In order to study the role played by RTP1 and RTP2 in regulating OR expression and trafficking in vivo, we consecutively knocked out these genes while the intervening ~500 kb genomic region was not disrupted in ES cells (*Figure 1A*). Following chimeric mice production and germline transmission, we established mouse lines with *Rtp1* and *Rtp2* double knock out alleles. We found no phenotypic difference between *Rtp1*(+/+);*Rtp2*(+/+) (wild-type) and *Rtp1*(+/−);*Rtp2*(+/−) (het) mice. The *Rtp1* (−/−);*Rtp2*(−/−) homozygous mutants (RTP1,2DKO) showed no gross defects outside the olfactory system. Heterozygous crosses gave rise to wild-type, heterozygous and homozygous adults in roughly 1:2:1 ratio (*Rtp1*(+/+);*Rtp2*(+/+) 19, *Rtp1*(+/−);*Rtp2*(+/−) 34, *Rtp1*(−/−);*Rtp2*(−/−) 15, n = 10 mating pairs), suggesting no embryonic or postnatal lethality. We validated the absence of RTP1 and RTP2 transcripts in RTP1,2DKO by performing RNA in situ hybridization (*Figure 1B*).

### RTP1,2DKO OSNs show defects in OR trafficking

It has been previously shown that RTP1 and RTP2 promote cell surface expression of ORs in the heterologous expression assays (*Saito et al., 2004*; *Zhuang and Matsunami, 2007*). Therefore, we used M71-IRES-tauGFP mice in which Olfr151 (also known as M71 and MOR171-2) expressing OSNs co-express tauGFP to examine the OSNs for OR trafficking defects (*Feinstein et al., 2004a*) (*Figure 1C*). In the RTP1,2DKO;M71-IRES-tauGFP OE, GFP staining was observed in the dendrites of Olfr151 positive OSNs (*Figure 1D*), indicating that the morphology of their OSNs remains unchanged. In contrast, immunostaining against Olfr151 (*Barnea et al., 2004*) was restricted to the cell body, indicating these OSNs are unable to traffic the OR to the dendrite (*Figure 1D*). Altogether, the data suggest that RTP1 and RTP2 are essential for OR trafficking.

### RTP1,2DKO mice have fewer mature sensory neurons

Upon examination of the OE, we found that its thickness was significantly reduced in RTP1,2DKO mice. (p=0.02 paired student t test) (*Figure 2A*). We therefore examined the expression of various OSN developmental markers and signaling molecules in the OE to evaluate areas occupied by mature and immature OSNs in RTP1,2DKO. We compared OMP and adenylate cyclase 3 (ACIII), markers for mature neurons (*Carter et al., 2004*; *Rogers et al., 1987*), in 21 day old RTP1,2DKO mice and their littermates (*Figure 2B*, *Figure 2—figure supplement 1*). We measured the area occupied by RNA in situ hybridization signal against OMP and found that mice showed an average of 22% reduction in RTP1,2DKO when compared to the wild-type (p<0.0001, paired student t test, wild-type mean area 71% ± 5 (SD), RTP1, 2DKO mean area 49% ± 4 (SD)) (*Figure 2C*, See methods for details). Comparison of the OMP positive layer from wild-type and RTP1,2DKO OE collected at 1-day-old, 21-day-old and 6-month-old mice showed a significant reduction in OMP expression at 1 day and 21 days (p=0.0003, Mann Whitney U test, p=0.0003, Mann Whitney U test) but not at 6 months (*Figure 2D*). Immunohistochemical analysis of expression of adenylate cyclase 3 (ACIII), a signaling molecule expressed in mature OSNs (*Wei et al., 1998*; *Wong et al., 2000*; *Col et al., 2007*) showed a 17% decrease in the area occupied by the staining in 21 day old RTP1,2DKO OE (p=0.0001, Mann Whitney U test, wild-type mean area 44% ± 9 (SD), RTP1,2DKO mean area 27%, ± 3 (SD)) (*Figure 2B*). Consistent with OMP expression, we observed a significant difference in ACIII expression at 1 day old (p=0.0079, Mann Whitney U test) but not at 6 months (*Figure 2E*).

GAP43 is a marker for immature olfactory neurons in the OE (*Meiri et al., 1988*; *Verhaagen et al., 1990*; *Treloar et al., 1999*) and area occupied by it shows a 7% increase in the area of the OE it occupies in 21-day-old RTP1,2DKO (p=0.03 Mann Whitney U test, wild-type mean area 20%, ± 5 (SD), RTP1,2DKO mean area 27%, ± 6 (SD)) (*Figure 2B*). No significant difference in the GAP43 positive layer is observed between RTP1,2DKO and their littermates at 1 day nor at 6 months (*Figure 2F*).

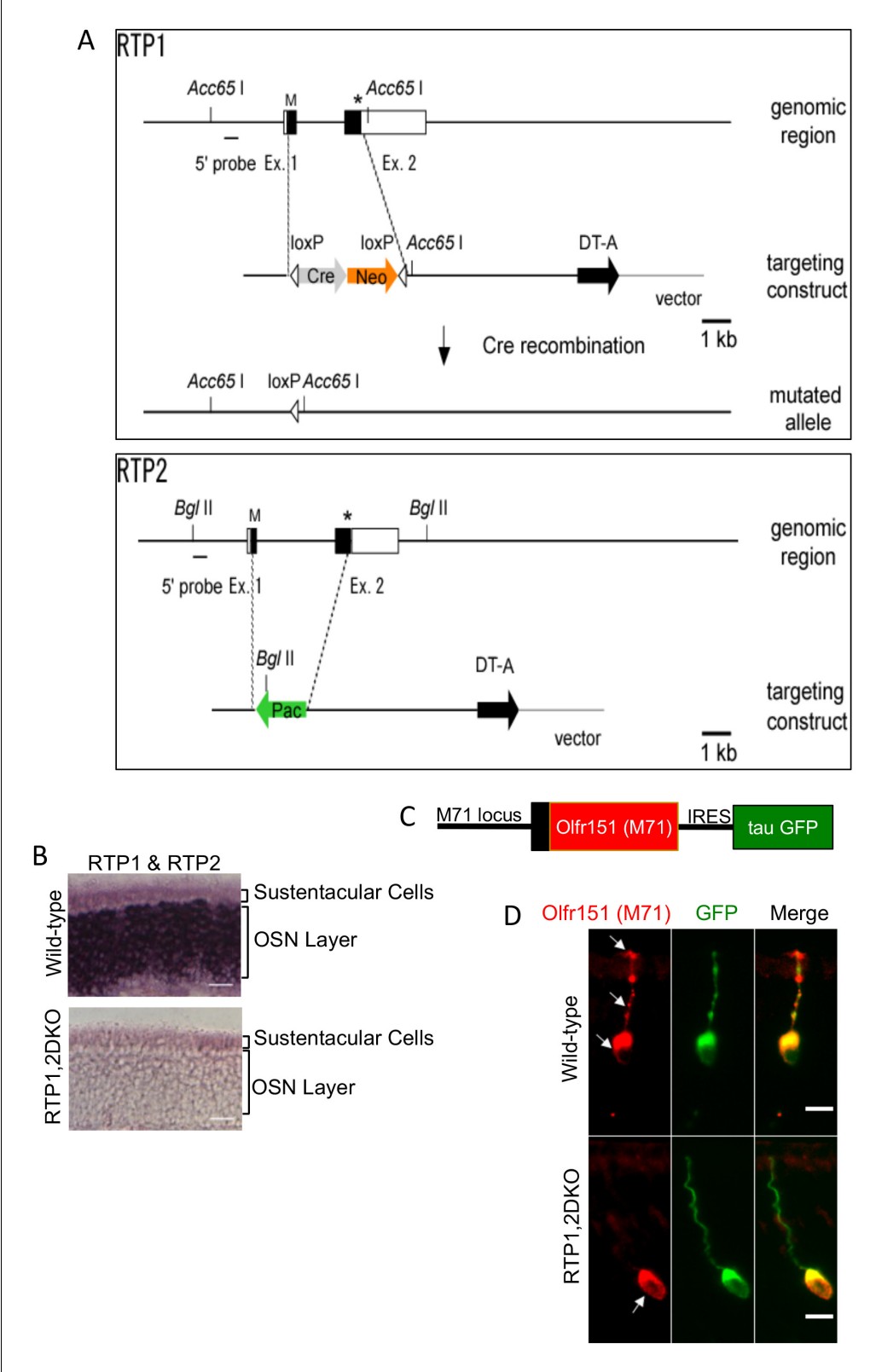

**Figure 1.** Deletion of RTP1 and RTP2 causes defects in the OE. (**A**) Strategy for knocking out RTP1 and RTP2 in series. (**B**) RNA in situ hybridization with probes specific to RTP1 and RTP2 in both wild-type and RTP1,2DKO showing that the knock out mice do not express either of these proteins. Scale bar = 50 μm. (**C**) Schematic depiction of M71-IRES-tau GFP. (**D**) Antibody against M71 (red) stains the dendrite in the wild-type OE (top) but not

*Figure 1 continued on next page*

Figure 1 continued

the RTP1,2DKO OE. On the other hand, the antibody against GFP shown in green stains the entire neuron from RTP1,2DKO;M71-IRES-tauGFP mice, which shows that M71 positive OSNs have dendrites.

## Odorant evoked electrophysiological responses in RTP1,2DKO mice are diminished

Upon observation of fewer OSNs in RTP1,2DKO mice and lack of OR trafficking to the cilia (*Figure 1D*), we sought to test the olfactory ability by electroolfactogram (EOG). We tested a diverse set of 7 odorants, in both wild-type and RTP1,2DKO littermates. Wild-type mice show robust EOG responses to all odorants at concentrations as low as 0.01% (*Figure 3A*). In contrast, RTP1,2DKO mice showed striking deficits in their response. Responses to most odors were identical to the blank stimulus (air only), although some sensitivity was maintained for a subset of odorants (2-heptanone, amyl acetate, isomenthone) compared to the wild-types (*Figure 3B–C*). Thus, the reduction of mature OSNs and the loss of surface OR expression corresponds to a dramatic loss of odorant sensitivity in RTP1,2DKO.

## OR expression is biased in RTP1,2DKO

To obtain a comprehensive view of gene expression changes in RTP1,2DKO, we performed an RNA-Seq on isolated whole olfactory mucosa including the OE and surrounding tissues. Differential expression analysis comparing RTP1,2DKO to wild-type littermates revealed that 3.8% of all genes (926/24,661) were differentially expressed between the two genotypes, among which 805 were downregulated and 121 were upregulated in RTP1,2DKO (FDR corrected $p < 0.05$, see Experimental Procedures for details). Canonical signaling molecules known to be expressed in mature OSNs including *Gnal* (Gαolf), *Adcy3* (ACIII), and *Cnga2* were less abundant in the RTP1,2DKO consistent with a reduced number of mature OSNs in absence of RTP1 and RTP2. We found no significant difference in the expression levels of housekeeping genes like *Gapdh* and *β actin* (*Supplementary file 1*) (*Kouadjo et al., 2007*), neither did we see any compensatory increase in other RTP family members *Rtp3* or *Rtp4*.

We then asked whether the loss of RTP1 and RTP2 equally affected all ORs. In a comparison between wild-type and RTP1,2DKO we found that 62% of intact ORs (678/1088) were significantly affected by the loss of RTP1 and RTP2 (*Figure 4A*). Close to half of the annotated intact ORs (562/1088) were downregulated in RTP1,2DKO (FDR corrected $p < 0.05$), consistent with fewer OSNs in the mutant. Unexpectedly however, a small subset of OR transcripts (116/1088) were upregulated in RTP1,2DKO mice (FDR corrected $p < 0.05$) (*Figure 4B*, *Figure 4—figure supplement 1(A-B)*).

The disparity in the abundance of transcripts for these ORs raised the possibility of a difference in probabilities of OSNs expressing each OR in RTP1,2DKO. To remove any possible confounding variables from non-OSN cells, we normalized our read counts using only reads mapped on intact ORs and found that 531/1088 were underrepresented and 202/1088 were overrepresented (FDR corrected $p < 0.05$) (*Figure 4C,F*, *Figure 4—figure supplement 1C*).

To further validate changes in numbers of OSNs expressing individual ORs in RTP1,2DKO mice, we carried out RNA in situ hybridization with probes against either underrepresented ORs (uORs) (*Figure 4C*) or overrepresented ORs (oORs) (*Figure 4F*). For all uORs tested, fewer OSNs were positive in RTP1,2DKO (*Figure 4D–E*) ($p < 0.05$ Mann-Whitney U test). In stark contrast, the frequencies for the tested oORs were greater in RTP1,2DKO ($p < 0.05$, Mann-Whitney U test) (*Figure 4G–H*). These results demonstrate that OR gene choice is biased in RTP1,2DKO towards a specific subset of receptor types.

Curiously, we found that oORs as a group are more abundantly expressed than uORs in the wild-type. The OR genes that were not classified as either underrepresented nor overrepresented (NS, not significant) exhibited a wide range of changes in expression levels between the wild-type and RTP1,2DKO, but are expressed at significantly lower abundance levels than both oORs and uORs (*Figure 4I*) ($p < 0.0001$ one-way ANOVA, Tukey's post hoc test).

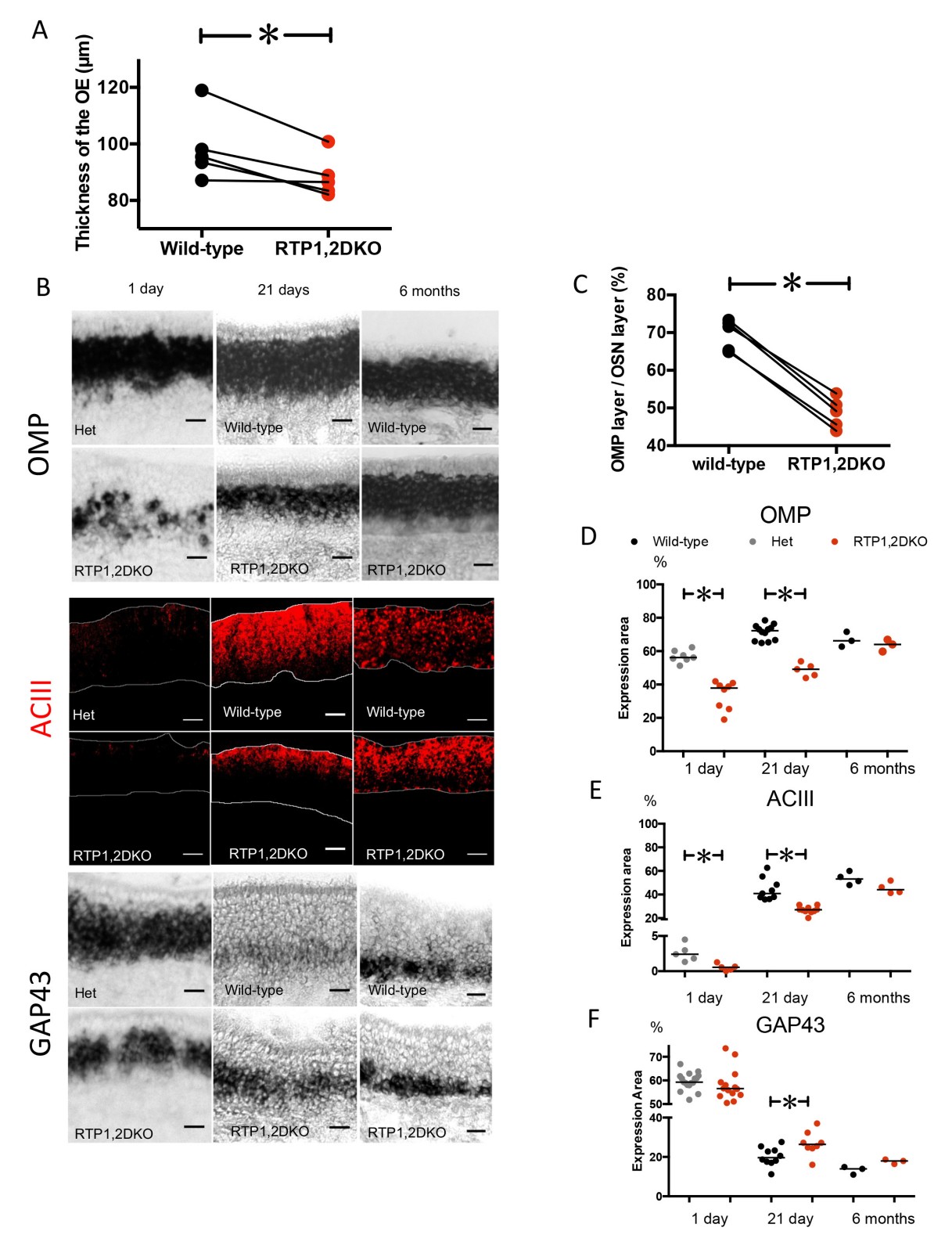

**Figure 2.** RTP1,2DKO mice have fewer mature sensory neurons. (**A**) Paired comparison of the thickness of the OE measured at five matched positions (see methods) between RTP1,2DKO and their wild-type littermate. (p=0.02, paired student t test). (**B**) RNA in situ hybridization against OMP (top), GAP43 (bottom) and IHC against ACIII (middle) at 1 day, 21 days and 6 months old. Scale bar = 25 µm. (**C**) Quantification of the percent area occupied by OMP RNA in situ hybridization signal from matched positions in the OE. Pair wise student t test shows a significant reduction in the area occupied

*Figure 2 continued on next page*

*Figure 2 continued*

by OMP staining in RTP1,2DKO. Error bars indicate SEM, p<0.0001, Paired student t test. (**D**) Comparison of percent area occupied by OMP between RTP1,2DKO and their het/wild-type littermates at different ages showing that RTP1,2DKO has fewer mature OSNs at 1 day (p=0.0003 Mann Whitney U test) and 21 day (p=0.0003 Mann Whitney U test) but there is no difference at 6 months (p=0.7, Mann Whitney U test). (**E**) Quantification of the area occupied by ACIII staining between RTP1,2DKO and their control genotype (hetetrozygous or wild-type) littermates at different ages showing that RTP1,2DKO has fewer mature OSNs at 1 day (p=0.0079 Mann Whitney U test) and 21 day (p<0.0001 Mann Whitney U test) but there is no significant difference at 6 months (p=0.1143, Mann Whitney U test). (**F**) Quantification of the area occupied by GAP43 staining between RTP1,2DKO and their het or wild-type littermates at different ages showing that RTP1,2DKO has more immature neurons at 21 day (p=0.0343 Mann Whitney U test).

The following source data and figure supplement are available for figure 2:

**Source data 1.** OE thickness and percent area occupied by the OMP layer, ACIII layer and GAP43 layer in the wild-type and RTP1,2DKO.

**Figure supplement 1.** Low magnification view of OMP in situ hybridization signals in OE sections.

## The proportion of OSNs expressing oORs increases in older RTP1,2DKO mice

We wondered what happens to the proportion of OSNs expressing uORs and oORs in RTP1,2DKO mice at different ages. We performed RNA in situ hybridization with a (1) a probe mix containing 11 uORs and (2) a probe mix containing 25 oORs, all expressed in the dorsal region of the OE on 1-day, 21-day and 6-month-old OE. In the case of the uOR mix, wild-type showed an increase for OSNs expressing the uORs we tested both at 21 days and 6 months (*Figure 5A–B*) consistent to the increasing proportion of mature OSNs in the OE indicated by larger OMP and ACIII layers (*Figure 2D–E*). However, RTP1,2DKO showed no obvious increase in the fraction of cells expressing these ORs with age, while on the other hand, in RTP1,2DKO the number of neurons expressing oORs showed a dramatic increase both from 1 day old to 21 days and from 21 days to 6 months (p<0.0001 one-way ANOVA, Tukey's post hoc test) (*Figure 5C–D*) demonstrating that the RTP1,2DKO OE is progressively populated by oORs.

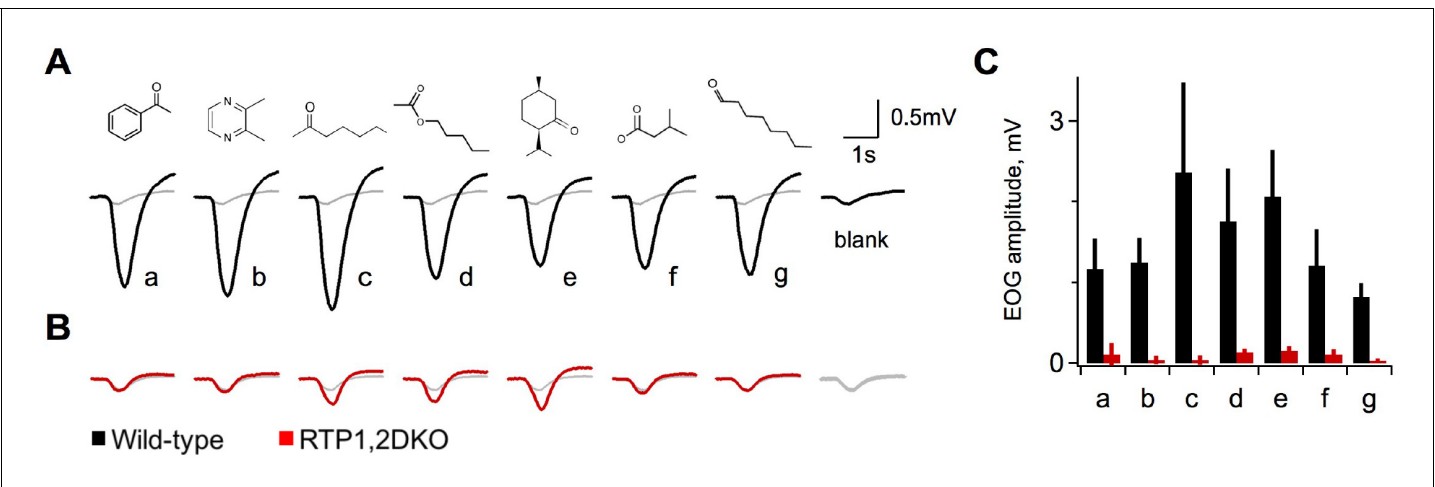

**Figure 3.** Diminished activity in response to odorants in RTP1,2DKO. (**A**) Electroolfactograms show the response to seven odorants wild-type. The grey line denotes the air only blank averaged over multiple interleaved trials interspersed within the series. (**B**) RTP1,2DKO responses to the same odors (**C**) Quantification of the EOG amplitudes for each of the seven odorants showing that only a few of the odors elicit responses from the RTP1,2DKO OE and these responses are lower than the wild-type. Each bar represents the difference between the peak of the odor minus the peak of the air only blank.

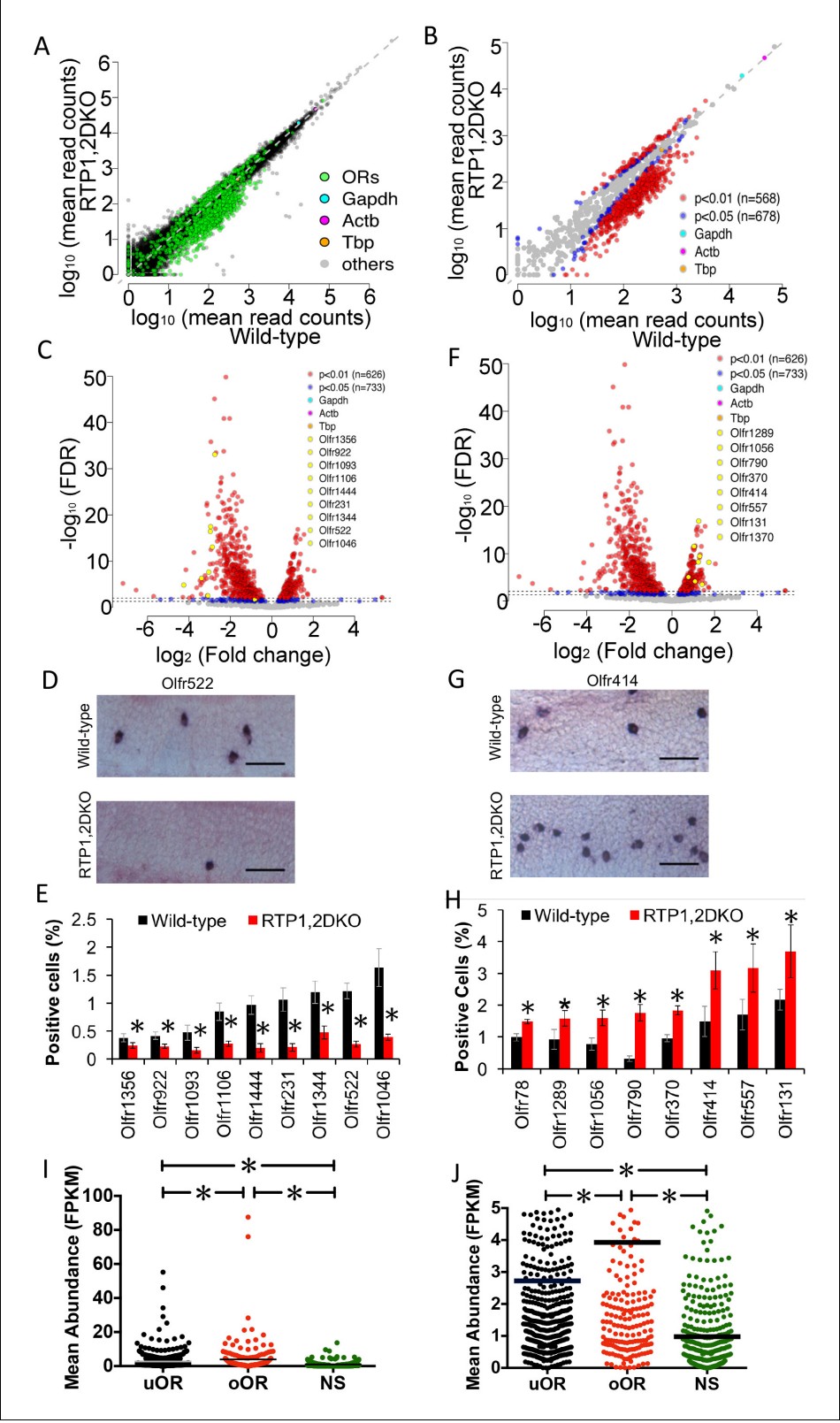

**Figure 4.** Representation of ORs in RTP1,2DKO. (**A**) Comparison of all transcripts between the wild-type and RTP1,2DKO, the green dots represent ORs, higher read counts for ORs are observed in the wild-type compared to RTP1,2DKO. (**B**) A comparison of the expression levels of ORs between the wild-type (x – axis) and RTP1,2DKO (y-axis). Red indicates uORs and oORs with p<0.01, blue indicates p<0.05. (**C**) A volcano plot showing the fold

*Figure 4 continued on next page*

*Figure 4 continued*

change of the expression levels (x-axis) of the ORs between wild-type and RTP1,2DKO using read counts normalized by OR genes. Red dots are ORs with p<0.01, blue: p<0.05, yellow dots signify candidate uORs chosen for validation. (D) Representative images for an in situ analysis with a probe specific to Olfr522 (uOR) where there are fewer positive OSNs in RTP1,2DKO when compared to the wild-type. Scale bar = 25 μm. (E) Quantification of the OSNs expressing uORs shown in (C); all the tested ORs showed smaller fractions of positive OSNs in RTP1,2DKO compared to the wild-type. p<0.05, Mann-Whitney U Test, n = 3 mice. (F) Volcano plot showing oORs with read counts normalized by OR genes. (G) Representative images for an in situ hybridization analysis with a probe specific to Olfr414 (oOR) where there are more positive OSNs in RTP1,2DKO when compared to the wild-type. Scale bar = 25 μm. (H) Quantification of the OSNs expressing oORs shown in (G); all the tested ORs showed greater fractions of positive OSNs in RTP1,2DKO compared to wild-type. p<0.05, Mann-Whitney U Test, n = 3 mice. (I) Plot of the mean abundance where each dot represents a single olfactory receptor classified as an uOR/ oOR/ NS based on normalization by ORs. The horizontal bars denote mean abundance (FPKM). oORs are significantly more abundant than uORs, NS are less abundant than both oORs and uORs (p<0.0001, one-way ANOVA, Tukey's post hoc test). (J) zoomed in view of the plot showing uOR/oOR and NS abundance, horizontal bars denote mean abundance (FPKM).

The following source data and figure supplement are available for figure 4:

**Source data 1.** Percent positive cell counts for the uORs and oORs in *Figure 4E* and 4 hr.

**Figure supplement 1.** Representation of ORs in RTP1,2DKO using all genes.

## Expression bias of an OR in RTP1,2DKO depends upon the protein sequence and not on the genomic location

We wondered whether the OR expression bias arose due to the effect of RTP on the regulatory elements of an OR's gene locus or the protein. In our initial investigation, we did not find an obvious pattern or clustering for the genomic locations nor did we find obvious conserved residues or motifs amongst uORs or oORs (*Figure 6A*, *Supplementary file 2*). In an attempt to causally identify the basis of the bias, we used a mouse expressing $\beta_2$AR-IRES-LacZ from the Olfr151 locus (*Feinstein et al., 2004b*) (*Figure 6B*) and asked whether the numbers of OSNs expressing Olfr151 or $\beta_2$AR are similarly affected in RTP1,2DKO. We chose Olfr151 as it is a uOR and $\beta_2$AR because it is a non-OR GPCR, capable of reaching the cell surface without the RTPs in heterologous culture and can replace a functional OR in native OSNs (*Omura et al., 2014*). We saw that fewer Olfr151 expressing OSNs were present in RTP1,2DKO (p=0.0028, Mann-Whitney U test), as expected for a uOR. Strikingly, more $\beta_2$AR positive OSNs were present in RTP1,2DKO compared with wild-type (p<0.001, Mann-Whitney U test) (*Figure 6C–D*), suggesting that it is the protein sequence and not the locus of the OR that determines whether a given OR is underrepresented or overrepresented.

Given that $\beta_2$AR is known to be efficiently trafficked to the cell surface when heterologously expressed in the absence of the RTPs, we asked whether uORs and oORs show differential capabilities in cell surface trafficking in heterologous cells. We carried out live cell surface staining of HEK293T cells transfected with either Rho tagged uORs or oORs in the absence of RTP1 and RTP2. In order to quantify the surface staining, we carried out FACS to measure the surface OR levels. oORs as a group showed more OR surface expression than uORs (p<0.05, Mann-Whitney U test) (*Figure 6E–G*). Notably, the ORs that show most robust cell surface expression were all oORs. Even though trafficking mechanisms between OSNs and HEK293T cells are likely to be different, our data are consistent with the idea that RTP-independent trafficking of ORs may be related to increased frequencies of OSNs expressing oORs.

## OSNs expressing oORs mature and function

Our results thus far suggest that OSNs expressing oORs are able to function despite the loss of the RTPs. In order to test this, we examined whether OSNs expressing uORs or oORs co-express OMP (*Figure 7A*). We found that the number of immature OMP-negative OSNs expressing uORs are similar in RTP1,2DKO and het controls, whereas the number of OMP-positive OSNs expressing uORs show a 69% decrease in RTP1,2DKO. (p=0.024 Mann Whitney U test) (*Figure 7B–D*).

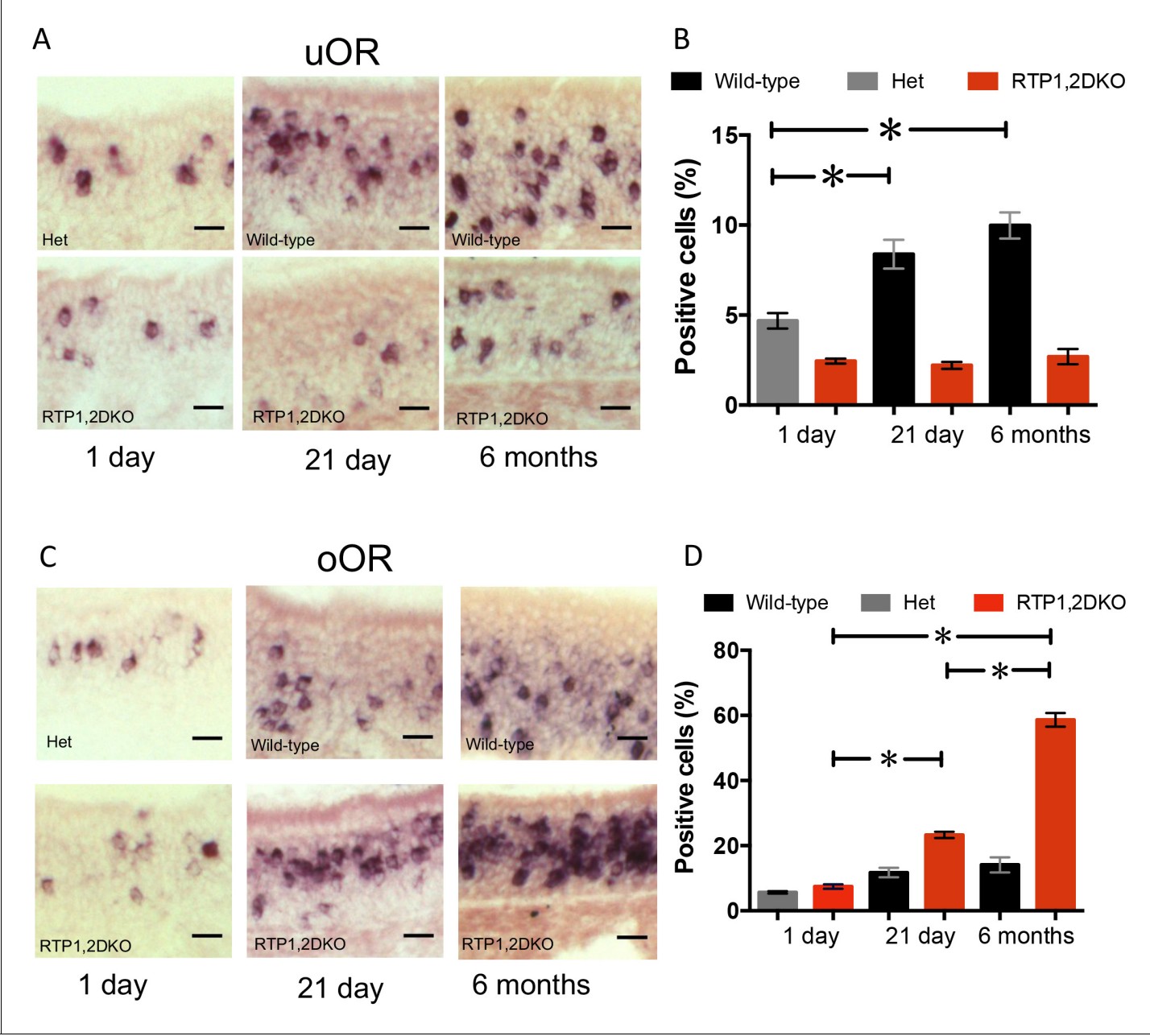

**Figure 5.** The proportion of OSNs expressing oORs increases in older RTP1,2DKO. (**A**) Representative images from 1 day, 21 day and 6 month OE stained with a probe mix against 11 of the uORs expressed in the dorsal OE. (**B**) Quantification of the percent dorsal uOR positive cells at different ages in RTP1,2DKO and their het or wild-type littermates. The fraction of cells positive for this probe significantly increases with age only in wild-type (p<0.0001 one-way ANOVA, Tukey's post hoc test). (**C**) Representative images from 1 day, 21 day and 6 month OE stained with a probe mix against 25 of the oORs expressed in the dorsal OE. (**D**) Quantification of the percent dorsal oOR positive cells at different ages between RTP1,2DKO and their het or wild-type littermates. The fraction of cells positive for this probe mix significantly increases with age in RTP1,2DKO (p<0.0001 one-way ANOVA, Tukey's post hoc test).

The following source data is available for figure 5:

**Source data 1.** Percent positive cell counts for the uOR and oOR probe mix at 1 day, 21 day and 6 month old OE.

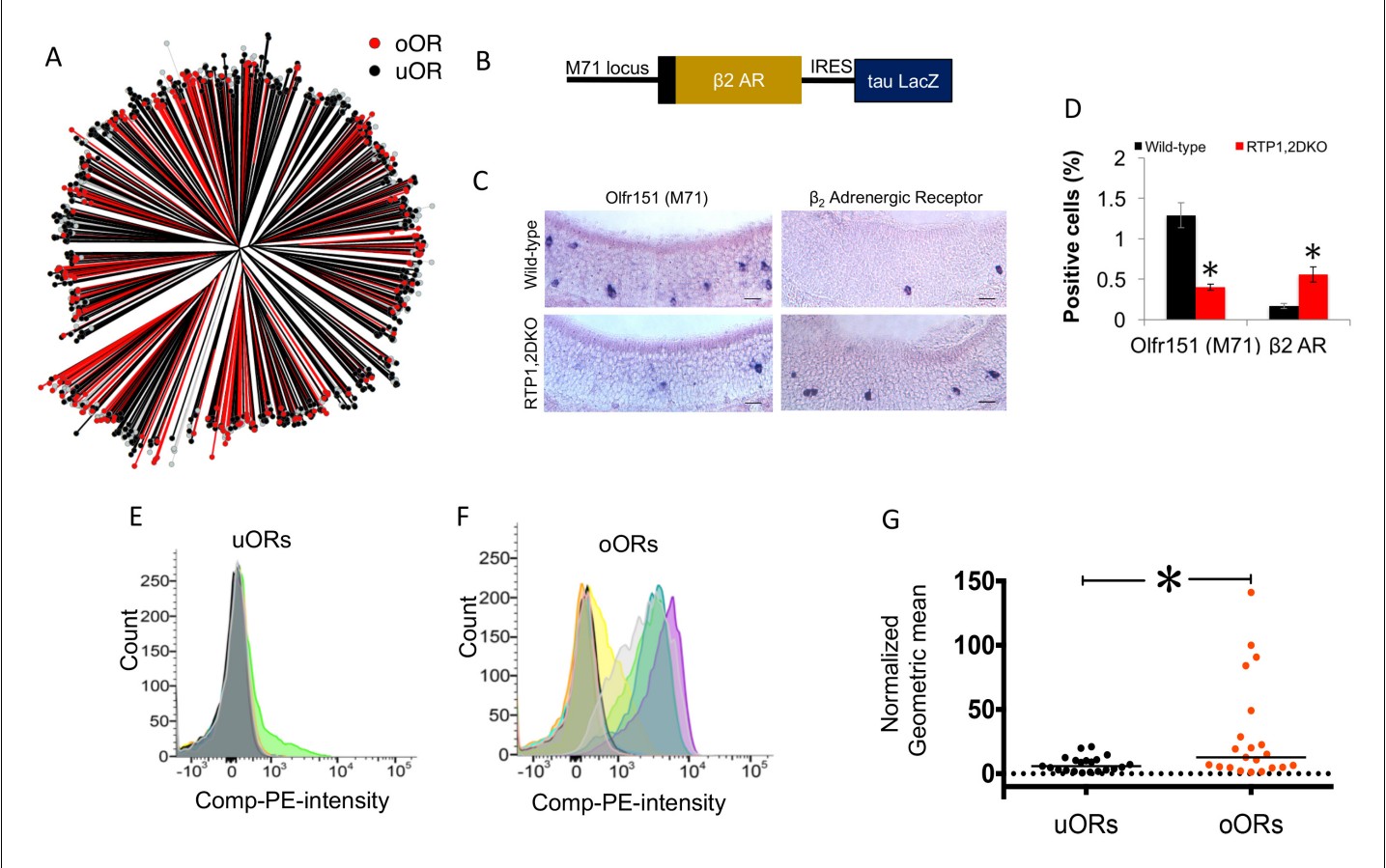

**Figure 6.** OR protein sequences determine representation in RTP1,2DKO. (**A**) Phylogenetic tree showing uORs in black and oORs in red. (**B**) Schematic depiction of $\beta_2$AR-IRES-tau LacZ. (**C**) The percent Olfr151 positive cells is smaller in RTP1,2DKO mouse (left panels). $\beta_2$Adrenergic receptor expressed from the Olfr151 locus shows more $\beta_2$AR cells in RTP1,2DKO mouse (right panels). (**D**) Quantification of the percent positive Olfr151 and $\beta_2$AR cells in wild-types vs RTP1,2DKO $p<0.05$ Mann-Whitney U test, n = 3 mice. (**E**) Representative FACS data graphing the number of cells (y-axis) vs the intensity of phycoerythrin staining expressing Rho tagged uORs (x-axis). Each color represents an individual uOR. (**F**) Representative FACS data graphing the number of cells (y-axis) vs the intensity of phycoerythrin staining expressing Rho tagged oORs (x-axis). Each color represents an individual oOR. (**G**) Comparison of the normalized geometric mean of the compensated PE intensity for all uORs vs all oORs tested. The geometric means are normalized to Olfr78 ($p=0.0483$ Mann-Whitney U test, uOR n = 23 genes, oOR n = 24 genes). Every geometric mean is calculated by counting the PE intensity across 10,000 cells.

The following source data is available for figure 6:

**Source data 1.** Normalized geometric mean for PE intensity obtained from our FACS experiment.

To evaluate the function of OSNs expressing uORs or oORs, we chose a uOR and an oORs that have been previously deorphanized. Olfr1395 is an oOR found to respond to 2,5-dihydro-2,4,5-trimethylthiazoline (TMT) and Olfr923, a uOR, to acetophenone in vivo (*Jiang et al., 2015*). Using the induction of phospho ribosomal protein S6 (pS6), a marker for neuronal activation (*Knight et al., 2012*), we found that OSNs expressing Olfr1395 in both het and RTP1,2DKO were activated to its cognate ligand TMT ($p<0.0001$ one-way ANOVA, Tukey's post hoc test) (*Figure 7E,G*). In contrast OSNs expressing Olfr923 in het, but not RTP1,2DKO were activated by their cognate ligand acetophenone (*Figure 7E–F*). These data show that OSNs expressing oORs mature and function in the RTP1,2DKO.

Immunostaining against active caspase 3, a cell death marker, suggested that RTP1,2DKO mice have an increased number of OSNs undergoing cell death ($p<0.01$, Mann-Whitney U test) (*Cowan et al., 2001*) (*Figure 7H–I*). Active caspase three staining in conjunction with OMP

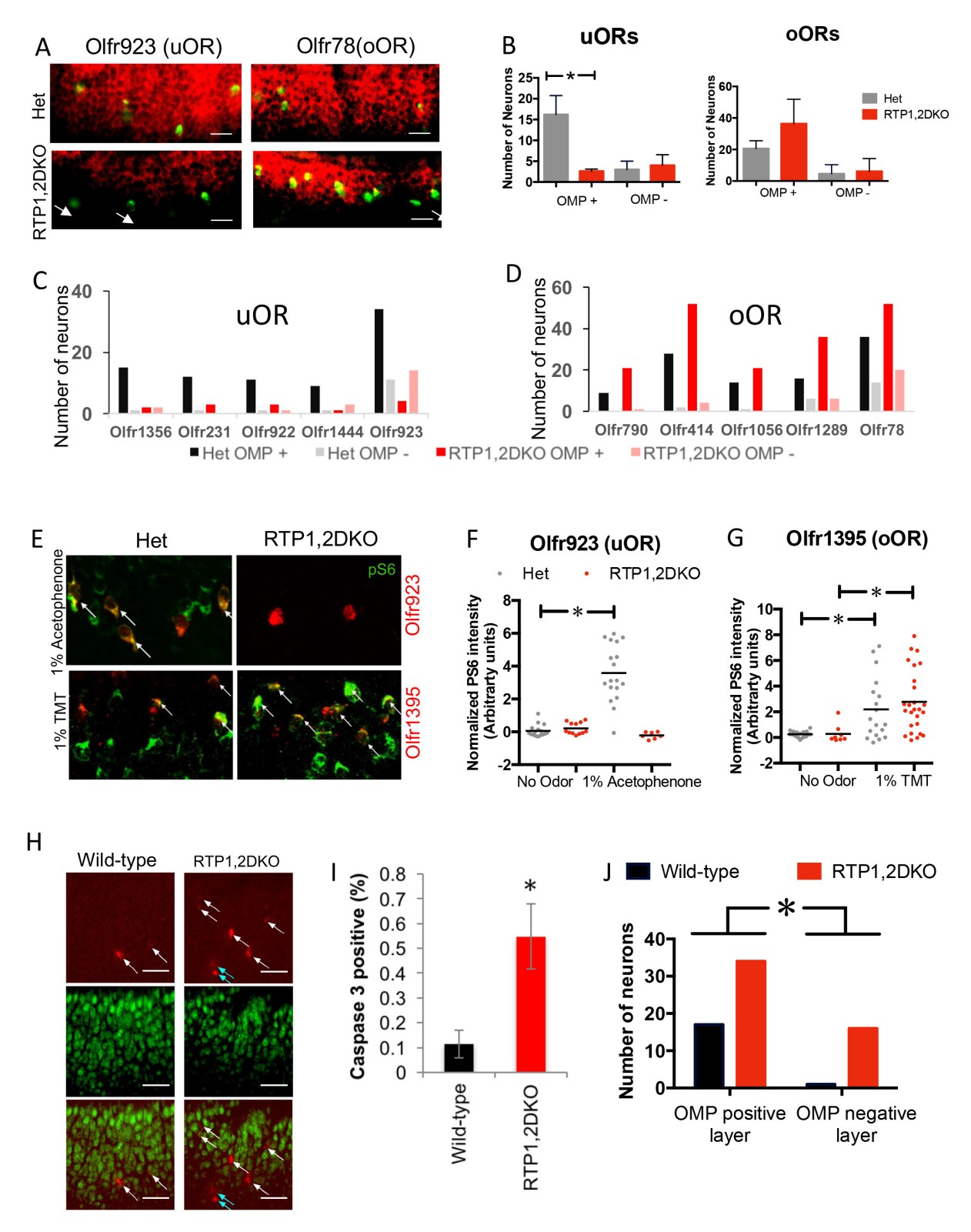

**Figure 7.** OSNs expressing oORs from RTP1,2DKO can mature and function. (**A**) Representative images showing the colocalization of Olfr923 (uOR) and Olfr78 (oOR) (green) with OMP (red) for het (top) and RTP1,2DKO (bottom). OMP negative OSNs are indicated with arrows. (**B**) Quantification of OMP positive OSNs for uORs and oORs as a group. For uORs there is a significant decrease (p=0.024 Mann Whitney U test, het n = 96, RTP1,2DKO n = 33) in the number of OMP positive OSNs in RTP1,2DKO whereas no significant difference is observed for oORs (p=0.4427 Fisher's exact test, het n = 126,
*Figure 7 continued on next page*

Figure 7 continued

RTP1,2DKO n = 213). (C) Quantification of the number of OSNs co-expressing OMP for individual uORs. (D) Quantification of the number of OSNs co-expressing OMP for individual oORs. (E) Representative images for pS6 staining (green) along with either a uOR (Olfr923) or an oOR (Olfr1395) in response to their cognate ligands. het (left) shows pS6 induction, whereas RTP1,2DKO (right) shows pS6 induction in response to an odor that stimulates the oOR but not the one that stimulates the uOR. All pS6 positive neurons are indicated by white arrows. (F) Quantification of the fold change in pS6 staining (pixel) intensity for Olfr923 positive cells in het (grey) and RTP1,2DKO mice (red) in response to 1% acetophenone. There is a significant increase in the pS6 induction in het (p<0.0001 one-way ANOVA, Tukey's post hoc test) but not RTP1,2DKO when subject to the odor. (G) Quantification of the fold change in pS6 staining pixel intensity for Olfr1395 positive cells in het (grey) and RTP1,2DKO mice (red) in response to 1% TMT. There is a significant increase in the pS6 induction in both het and RTP1,2DKO when subject to the odor (p=0.0002, one-way ANOVA, Tukey's post hoc test). (H) Wild-type (left) and RTP1,2DKO OE (right) stained with an antibody against the active form of caspase3 (red) and OMP (green). The white arrows indicate cells expressing active caspase 3 in the OMP positive layer and the blue arrows show the active caspase 3 positive cells outside the OMP positive layer. Scale bar = 50 µm. (I) Quantification of the percentage of active caspase3 positive cells. Error bars indicate SEM, p<0.01, Mann-Whitney U Test (n = 3 mice) (J) Quantification of OMP and active caspase 3 double staining showing that RTP1,2DKO have significantly more OSNs undergoing cell death both in mature and immature OSN layers compared to wild-type. (p=0.029, Fisher's exact test)

The following source data is available for figure 7:

Source data 1. Numbers of uOR and oOR neurons found within the OMP layer and outside it.
Source data 2. Normalized pS6 staining intensity for Olfr923 and Olfr1395 positive cells from het and RTP1,2DKO OE in response to 1%acetophenone and 1%TMT respectively.

suggested that more OSNs in immature and mature layers both undergo cell death in RTP1,2DKO (*Figure 7H*). Notably, immature OSNs in the OMP negative layer rarely undergo cell death in wild-type (*Figure 7J*) (*Jia et al., 2010*). Together, these observations are consistent with the idea that OSNs expressing uORs are more likely to undergo cell death in RTP1,2DKO.

## Persistent expression of nATF5 is observed in RTP1,2DKO OSNs expressing uORs

The above findings reveal the expression of a biased OR repertoire in RTP1,2DKO, with about a half of the ORs being underrepresented but a small subset of ORs overrepresented. Previous studies have shown that the unfolded protein response (UPR) plays an important role in OR gene choice mechanisms. During the initial phase of OR expression, the UPR pathway gives rise to an increased translation of nuclear activating transcription factor (nATF5) over an upstream inhibitory ORF via eIF2α signaling (*Godin et al., 2016*). Once the OSN has matured, OR gene choice is stabilized and the UPR is relieved (*Dalton et al., 2013*). We hypothesized that the lack of RTP1 and RTP2 causes persistent UPR in OSNs expressing uORs, leading to unstable OR gene choice in these OSNs, which in turn contributes to the skewed OR repertoire in mice lacking RTP1 and RTP2. We first asked whether the expression of nATF5, a marker for UPR, is different in RTP1,2DKO. Indeed, we observed that there were more nATF5 positive OSNs in RTP1,2DKO mice (p<0.001, Mann-Whitney U Test) and some of these OSNs were located closer to the apical surface, a phenomenon that was not observed in the wild-types, suggesting that nATF5 expression persists during the OSN development in RTP1,2DKO animals (*Figure 8A–B*). Similarly, expanded expression was observed for LSD1, a histone demethylase whose expression depends on nATF5 (*Figure 8C*). Next, we asked whether the ectopic expression of nATF5 in RTP1,2DKO is due to their protein sequence or the gene locus. We first compared the co-expression of nATF5 and Olfr151 between the wild-type and RTP1,2DKO using M71-IRES-tauGFP mice, and then asked whether the same co-localization occurs for $\beta_2$AR expressed from the Olfr151 gene locus using $\beta_2$AR-IRES-LacZ mice. We observed that the number of OSNs co-expressing Olfr151 and nATF5 was significantly higher in RTP1,2DKO than in the wild-type (p<0.05, Fisher's exact test). In contrast, the number of OSNs co-expressing $\beta_2$AR and nATF5 in the $\beta_2$AR:IRES:tauLacZ mouse was not different from the wild-type (p=1, Fisher's exact test) (*Figure 8D–F*). As Olfr151 but not $\beta_2$ARs require RTPs for surface expression, these data suggest that delivery of ORs to the membrane plays a role in terminating the UPR.

We next asked whether the expression of nATF5 in RTP1,2DKO mice was different in OSNs expressing uORs or oORs. In order to answer this question, we carried out fluorescent in situ hybridization against 7 uORs and 6 oORs along with immunohistochemistry for nATF5 and quantified

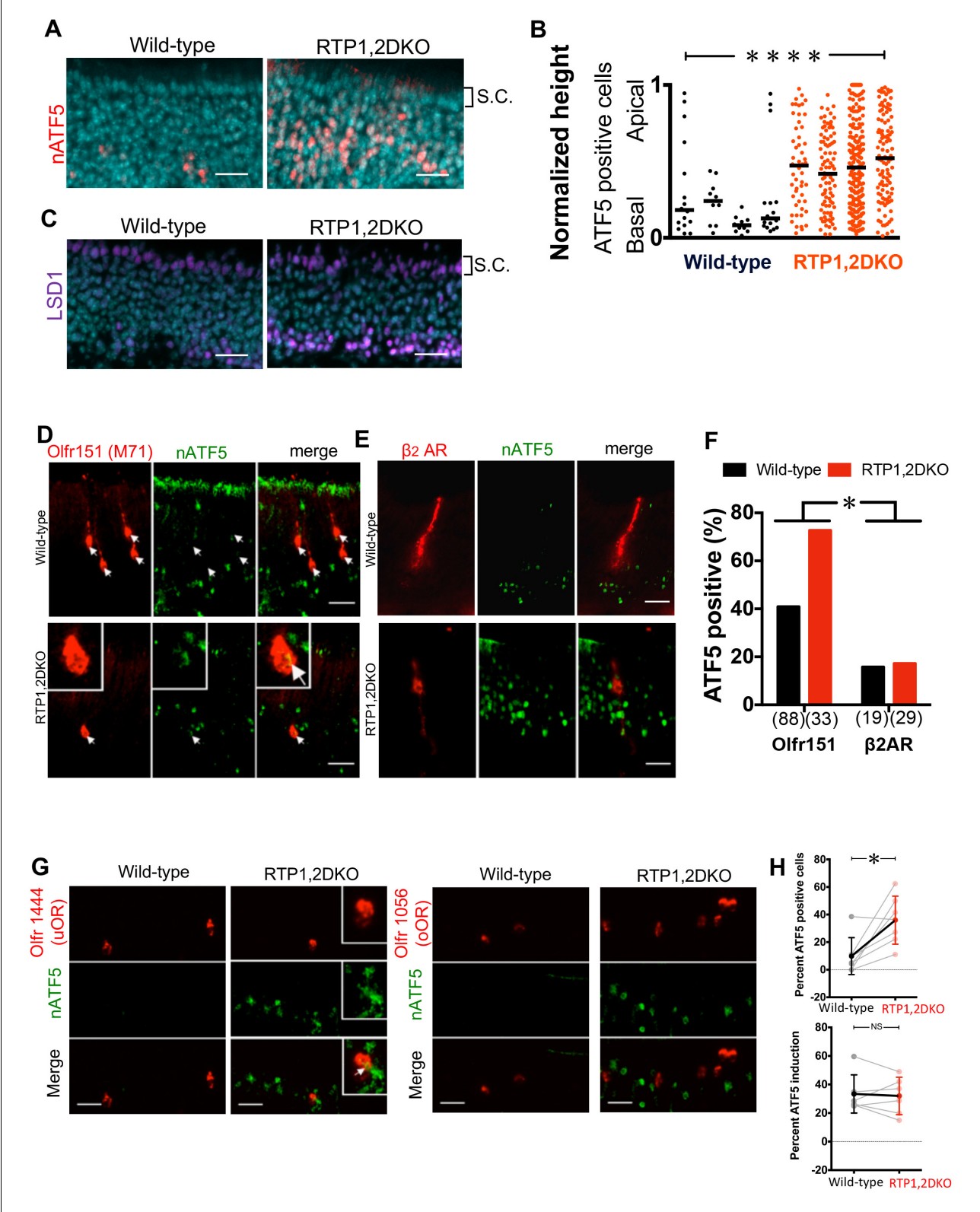

**Figure 8.** nATF5 expression persists in OSNs expressing uORs but not oORs in RTP1,2DKO. (**A**) Expanded expression pattern of nATF5 is observed for RTP1,2DKO Scale bars = 25 μm. S.C. = Sustentacular cells. (**B**) Analysis of individual sections of wild-type and RTP1,2DKO OE for ATF5 expression, RTP1,2DKO mice have a larger number of nATF5 positive cells and they are more apically situated compared to wild-type p=0.0049, Mann-Whitney U test, n = 3 mice. (**C**) Similar expanded expression pattern is observed for LSD1. Scale bars = 25 μm. S.C. = Sustentacular cells. (**D**) Representative

*Figure 8 continued on next page*

*Figure 8 continued*

images of Olfr151 and nATF5 in wild-type vs RTP1,2DKO OE. Inset: higher magnification, arrow head: nATF5 and Olfr151 co-localization. (E) Representative images from $\beta_2$Adrenergic receptor IRES tauLacZ ($\beta_2$AR) with antibody staining against LacZ indicating $\beta_2$AR positive OSNs and nATF5. (F) Quantification of the number of OSNs positive for both Olfr151 and nATF5 (left) and $\beta_2$AR and nATF5 (right). In RTP1,2DKO, there was a significant increase in the number of nATF5-Olfr151 double positive OSNs p=0.0022, Fisher's exact test, n = 3 mice but not $\beta_2$AR and nATF5 double positive neurons. p=1, Fisher's exact test, n = 3 mice. (G) Left: Representative images for Olfr1444 (uOR) (red) co-localization with nATF5 (green) signal in wild-type OE (left) and RTP1,2DKO (right). The inset shows a higher magnification of a single OSN positive for Olfr1444 and nATF5. Right: Representative images for Olfr1056 (oOR) shown in red. (H) Top: Percent nATF5 positive OSNs expressing 7 uORs in the wild-type vs the RTP1,2DKO. The solid points indicate the overall mean of all uORs and the solid line shows that more RTP1,2DKO OSNs expressing uORs co-localize with nATF5. p=0.007, Mann-Whitney U test, n = 2 mice. Bottom: Percent nATF5 positive OSNs expressing 6 oORs in the wild-type vs the RTP1,2DKO. (p=0.937, Mann-Whitney U test).

The following source data is available for figure 8:

**Source data 1.** Percent uOR/oOR positive cells that co-localize with nATF5 in wild-types and RTP1,2DKO.

colocalization of the OR signal with nATF5 staining. As expected, higher numbers of OSNs expressing uORs colocalized with nATF5 in RTP1,2DKO as a group (p=0.007, Mann-Whitney U test), whereas no significant difference was observed when oORs were tested (p=0.937, Mann-Whitney U test) (*Figure 8G–H*). These observations suggest that OSNs expressing uORs contribute to the expanded expression of nATF5 in RTP1,2DKO and are further consistent with the idea that surface trafficking of ORs is linked to turning off UPR.

## OR gene expression is unstable in OSNs expressing Olfr151 in RTP1,2DKO

Increased nATF5 levels in uOR-expressing OSNs of RTP1,2DKO mice suggests a lack of stable OR gene choice in these neurons. This led us to hypothesize that OSNs that initially express a uOR may later turn off the OR and stabilize the expression of another OR. To directly address this, we used a lineage tracing strategy to study the stability of OR gene choice in RTP1,2DKO (*Shykind et al., 2004*; *Dalton et al., 2013*; *Abdus-Saboor et al., 2016*). In our assay (*Figure 9A*), we crossed a mouse that had one allele expressing Cre recombinase under the Olfr151 promoter, M71-IRES-Cre, to a mouse that had the Cre inducible fluorescent reporter Rosa26-lox-stop-lox-tdTomato (*Madisen et al., 2010*). In this mouse, any OSN expressing the M71-IRES-Cre allele at any point in time, would give rise to the permanent expression of tdTomato even if the OSN went on to express a different OR gene. We counted the number of tdTomato positive neurons that were also Olfr151 positive (double positive), which reflect the OSNs that initially chose as well as stably express Olfr151. OSNs that were tdTomato positive but not Olfr151 positive are the ones that switched their initial gene choice. We found that only 17% of tdTomato positive cells from RTP1,2DKO OE (n = 24/140 neurons) also express Olfr151 in comparison to 31% OSNs from the wild-type (n = 79/258 neurons) (p<0.05, Fisher's exact test) and 38% (n = 66/172 neurons) in the heterozygotes (p<0.05, Fisher's exact test) (*Figure 9B–C*). This suggests that the loss of RTPs leads to the frequent termination of Olfr151 gene expression.

To determine whether the Olfr151 gene switched to an oOR within the same locus we carried out a co-localization analysis between Olfr143, an oOR within the Olfr151 locus and tdTomato under the control of M71-cre (*Figure 9—figure supplement 1*). We found no Olfr143 and tdTomato double positive OSNs in both RTP1,2DKO or their wild-type littermates, indicating that there is no higher likelihood of the gene switching between these ORs.

## Olfr151 glomeruli are not formed in RTP1,2DKO

Lastly, we investigated OSN axon targeting to the olfactory bulb (OB) in RTP1,2DKO. Olfactory axons entered the OB and innervate to the glomerular layer based on our OMP immunostaining in RTP1,2DKO (*Figure 10A*). However, using M71-IRES-tau GFP mice we found that Olfr151 expressing OSNs were unable to converge in the OB in RTP1,2DKO while their wild-type littermates had two Olfr151 glomeruli in each of their OBs as expected (*Figure 10B–C*). To investigate whether the axon targeting defect was ubiquitous to all receptors, we used $\beta_2$AR-IRES-tauLacZ (*Feinstein et al.,*

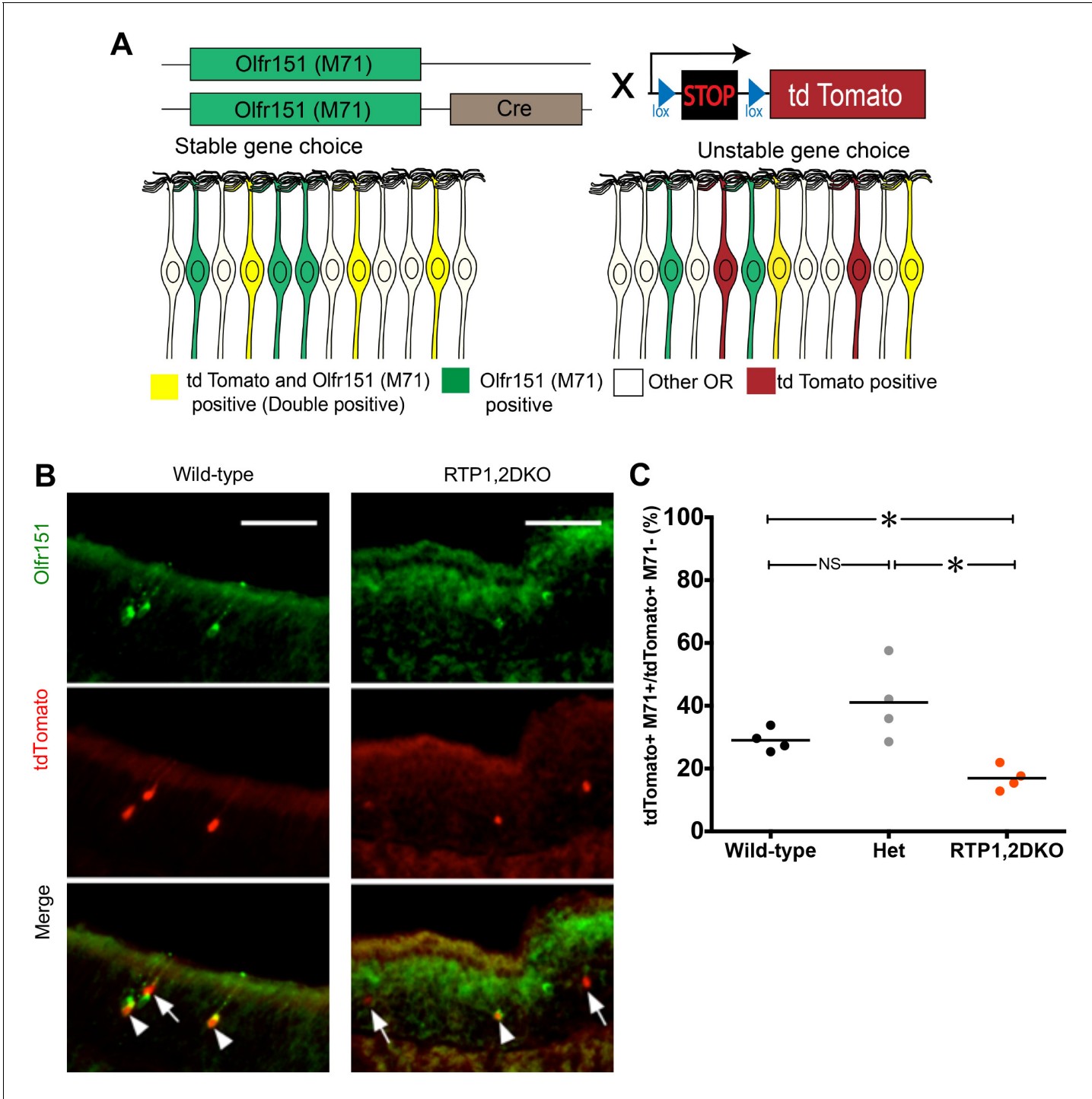

**Figure 9.** Unstable expressing of M71 in RTP1,2DKO. (**A**) Schematic depiction of OSN lineage tracing. We crossed a mouse carrying M71–IRES- Cre with Rosa26-lox-stop-lox-tdTomato. In the progeny, the expression of Olfr151 (M71) in an OSN will drive the expression of Cre, leading to the permanent production of tdTomato by the removal of the transcriptional stop sequence. Larger numbers of tdTomato positive OSNs that do not express Olfr151 would indicate unstable gene expression (right). On the other hand, if tdTomato OSNs largely stained positive for Olfr151 (double positive, shown in yellow), it would indicate stable OR expression. (**B**) Representative images from the wild-type and RTP1,2DKO OE stained with antibody against Olfr151 (green) and tdTomato (red). Arrow heads indicate double positive OSNs and the arrows show tdTomato positive and Olfr151 negative OSNs. (**C**) Each point represents the ratio of the number of tdTomato and Olfr151 double positive OSNs to the number of only tdTomato positive OSNs in one mouse. RTP1,2DKO mice have significantly lower OR gene choice stability compared to both wild-type and het mice. p<0.05, , Fisher's Exact test, n = 4 mice.

*Figure 9 continued on next page*

*Figure 9 continued*

The following figure supplement is available for figure 9:

**Figure supplement 1.** OSNs expressing Olfr151 do not switch to Olfr143.

*2004b*) (*Figure 6B*). Both the wild-type and RTP1,2DKO mice formed glomeruli, however the mutant mice had ectopic glomeruli for OSNs expressing this GPCR (*Figure 10B–C*). These data suggest the RTP1,2DKO mice do not have a complete set of glomeruli, but retain the ability to form them. We observed tdTomato-positive axons forming small glomeruli in RTP1,2DKO;M71-IRES-cre; Rosa26-lox-stop-lox-tdTomato mice (2 out of 3 mice examined had 1 glomerulus each)(*Figure 10B–C*). This may indicate that OSNs initially expressing Olfr151 switch and/or stabilize expression of a specific OR, presumably an oOR, and axons from these OSNs can converge in the OB.

## Discussion

In the current study, we investigated the in vivo role of RTP1 and RTP2 in the mouse olfactory system by generating and analyzing RTP1,2DKO mice. Our results demonstrate a surprising link between receptor trafficking of ORs, the UPR and OR gene choice.

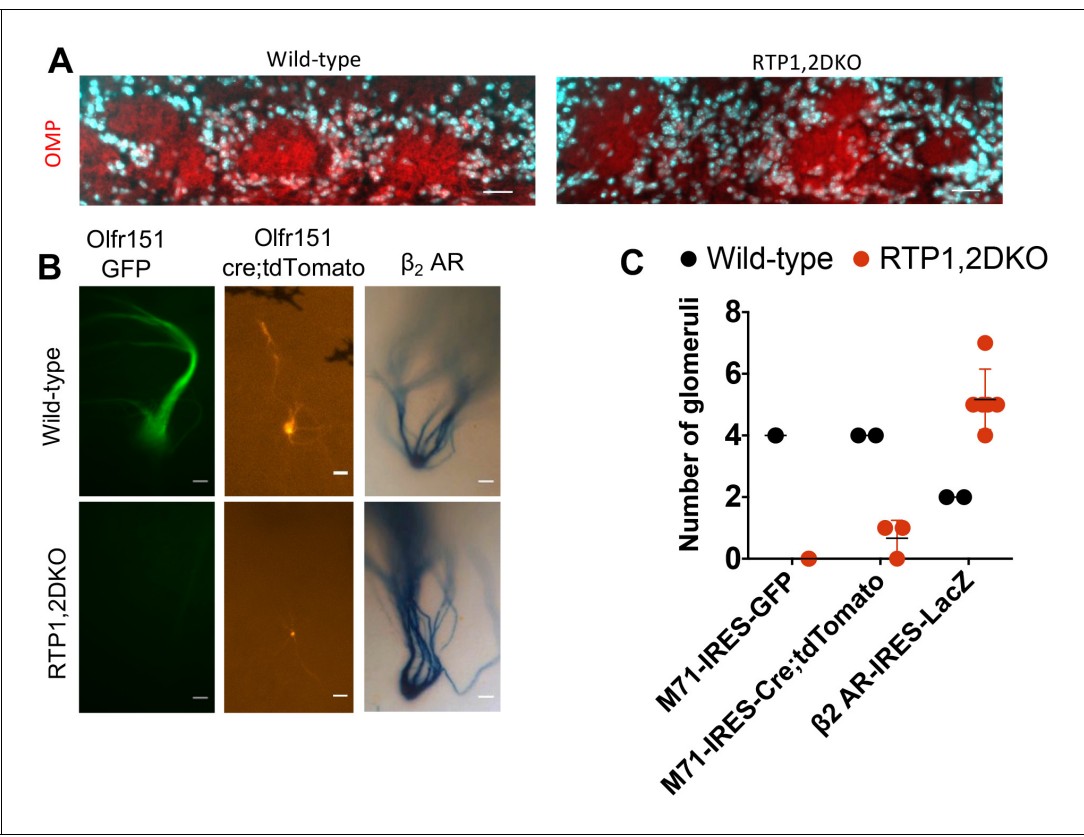

**Figure 10.** RTP1,2DKO mice are able to form glomeruli. (A) OMP staining shown in red and nuclear staining in cyan. Both wild-type and RTP1,2DKO mice have OMP in the glomerular layer. Scale bar = 25 µm. (B) A whole mount GFP fluorescence from axons expressing M71 from M71-IRES-tauGFP mice (left), tdTomato fluorescence from M71-IRES-Cre; Rosa26-lox-stop-lox-tdTomato (middle) and LacZ positive axons from $\beta_2$AR-IRES-tauLacZ mice (right). RTP1,2DKO OBs lack Olfr151 (M71) glomeruli but have tdTomato and LacZ positive ones, while labeled glomeruli are observed in wild-type with M71-IRES-tauGFP, M71-IRES-Cre; Rosa26-lox-stop-lox-tdTomato and $\beta_2$AR-IRES-tauLacZ. Only the dorso-lateral OB are visible for $\beta_2$AR-IRES-LacZ in our preparation. Scale bar = 25 µm. (C) Quantification of the total number of glomeruli observed in wild-type and RTP1,2DKO OBs. Each dot represents one mouse.

### Differential control of OR representations by RTPs

The absence of RTP1 and RTP2 leads to the underrepresentation of nearly half of the ORs, while about 10% of the ORs are significantly overrepresented. This translates into a change in the numbers of OSNs choosing each OR. How do the RTPs regulate the probability of OR gene choice?

Our data indicate that protein sequences of ORs differentially influence OR gene choice, which is linked to trafficking of ORs to the cell surface. Our attempts to identify protein motifs, domains, or features specific to either uORs or oORs were so far unsuccessful. This fits with recent reports where large-scale mutational analysis of Olfr151 in heterologous cells failed to identify any specific amino acids or domains that regulate its cell surface expression (*Hague et al., 2004*; *Jamet et al., 2015*). Nevertheless, this study provides a large set of sequence information of ORs that will allow us to conduct future structure-function studies by testing uOR/oOR chimeras and/or searching for hidden features within uORs or oORs, which may in turn give us clues as to why nearly half of the ORs are retained in the ER in the absence of the RTPs.

### Prolonged UPRs in OSNs expressing uORs in RTP1,2DKO

Previous studies have suggested the UPR protein nATF5 is induced once an OSN starts actively expressing an OR and is lost when OR expression is stabilized (*Dalton et al., 2013*). Our results show an expanded expression of nATF5 in RTP1,2DKO OSNs, suggesting persistent UPR during OR gene choice. Importantly, the frequency of co-localization between nATF5 and uORs increases in RTP1,2DKO whereas this increase is not observed for oORs. This suggests that the persistent UPR phenotype observed in RTP1,2DKO is due to the OSNs that express uORs resulting in unstable OR gene choice for these OSNs. This is reinforced by our observation that OSNs initially expressing Olfr151, a uOR, are more likely to terminate their expression in RTP1,2DKO mice. We present a

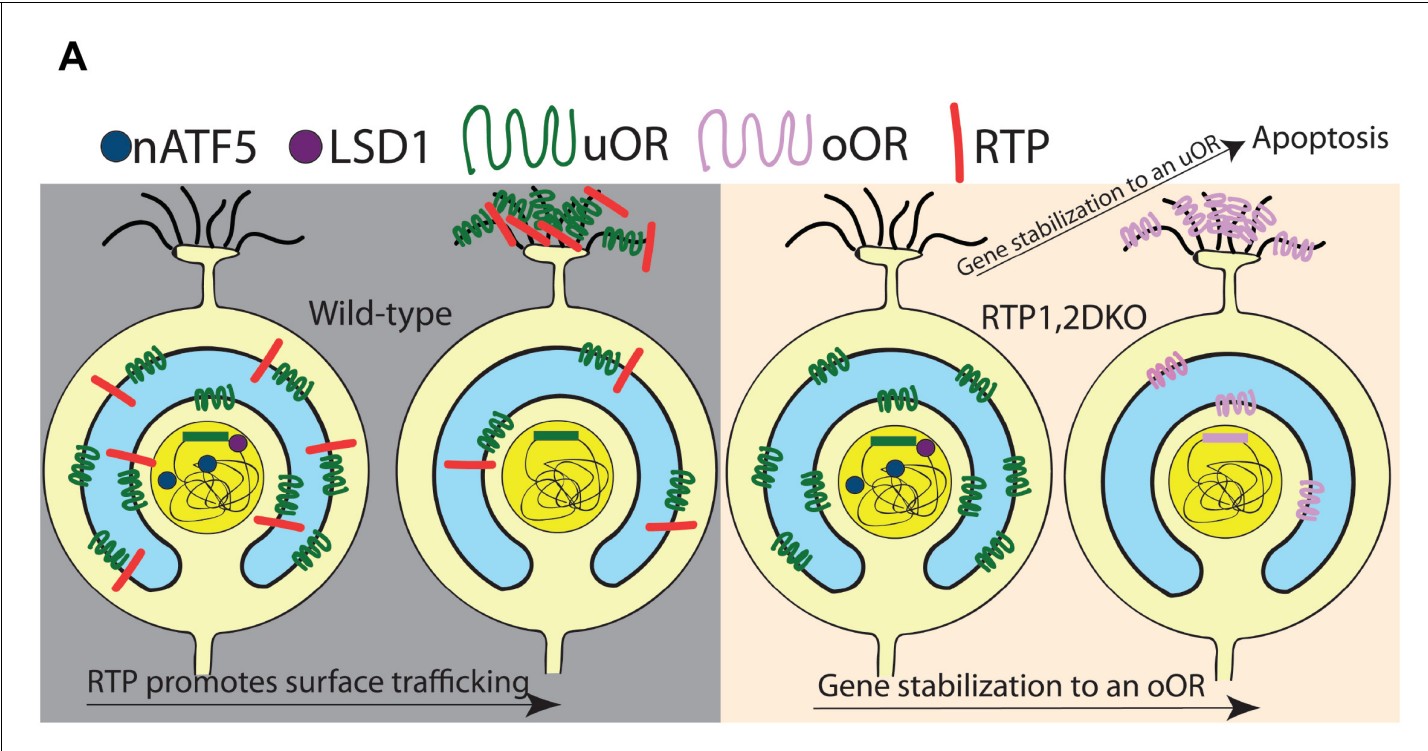

**Figure 11.** Model for the role of RTP1 and RTP2 in OR gene choice. (**A**) A model showing that in the wild-type (left), ORs are transported to the surface of the cell in conjunction with the RTPs. In the absence of the RTPs (right), oORs reach the surface of the cell and these OSNs survive, whereas uORs are not trafficked to the surface and OSNs expressing them show a persistent nATF5 expression. These uOR-expressing OSNs undergo OR choice switching and the OSNs that switch to an oOR are able to survive, leading to oOR overrepresentation. Those that are unable to switch to an oOR undergo cell death.

model for the role played by RTP1 and RTP2 in the gene choice made by an OSN (*Figure 11A*). In our model, the RTPs suppress UPR response by allowing ORs to exit the ER and be transported to the plasma membrane. uORs are not trafficked to the cell surface in the absence of the RTPs, giving rise to persistent UPR in the OSNs that express them; as a consequence, destabilization of the initial OR gene choice leads to cell death or to the stabilization of another OR. In contrast, oORs are trafficked to the cell surface as functional proteins in the absence of the RTPs, allowing these OSNs to terminate UPR and stabilize gene expression. Although we cannot rule out that RTPs themselves play a role in the elimination of the UPR response, the lack of increase in nATF5 observed in OSNs expressing oORs in RTP1,2DKO suggests interaction between the RTPs and the UPR pathway through ORs. Even for OSNs expressing oORs and $\beta_2$AR, UPR is likely to be induced in the initial stage (*Dalton et al., 2013*). Recent reports utilizing single cell RNA-Seq suggest that immature OSNs express as many as 12 ORs at low levels but mature OSNs only show the expression of one dominant OR (*Saraiva et al., 2015*; *Tan et al., 2015*; *Hanchate et al., 2015*). Low-level expression of uORs precedes the expression of the RTPs and may be sufficient to trigger UPR in the developing OSNs. Alternatively, both oORs and uORs induce UPR at the initial stage.

RNA-Seq data in the wild-type suggests that oORs as a group tend to be more frequently chosen. It could be that initial expression of RTP1 and RTP2 in the developing OSNs is not stable or abundant enough, causing oORs to be stabilized. Alternatively oORs tend to 'win the competition' (*Nguyen et al., 2007*; *Abdus-Saboor et al., 2016*) even in the presence of RTPs probably because of its ability to suppress UPR. Both change in probability of OR gene choice and biased cell death could alter OR population representation both in the level of transcripts and in the number of OSNs. The relative contributions of cell death and gene switching to the differential representation of ORs can be further clarified using Bax knockout mice where cell death in developing neurons is suppressed (*Robinson et al., 2003*).

OR genes that did not significantly change in RTP1,2DKO mice showed lower abundance in wild-type mice suggesting that these ORs are chosen less frequently. Deeper sequencing and/or increased sample sizes will help classify these ORs as underrepresented, overrepresented, or not changed.

In our lineage tracing experiment, we were unable to distinguish tdTomato-positive, Olfr151-negative OSNs between those express Olfr151 as one of the low-level ORs, and the others express Olfr151 as the dominant OR later switch to express another OR. Irrespective of this our data show greater gene instability for Olfr151 expression without the RTPs. Consistent with multiple OR gene expression in developing OSNs, our lineage tracing results suggest that only 31% of OSNs that initially expressed Olfr151 went on to stabilize its expression in the wild-type. Curiously, the vast majority of OSNs that initially expressed Olfr1507 (MOR28) show stable expression in a similar experiment (*Shykind et al., 2004*; *Dalton et al., 2013*), suggesting that some ORs are more likely to be stabilized than other ORs.

The frequency with which any OR is chosen is different and the underlying cause for this difference remain unknown. Our results show that the number of OSNs expressing $\beta_2$AR expressed from the Olfr151 locus is lower than the number expressing Olfr151. Our data call for future experiments to test whether protein sequences of ORs differentially influence initial OR gene choice.

## RTP1,2DKO mice show diminished but not abolished responses to odors

Our data suggest that mice without RTP1 and RTP2 had diminished but not abolished responses to odorants. Even though we see a clear reduction in the number of mature OSNs and dramatically diminished responses to odors in RTP1,2DKO mice, functional OSNs are not completely eliminated. Previous studies suggest that anosmic mice often die in the first few days after birth (*Brunet et al., 1996*), but the RTP1,2DKO mice seem to have the sufficient olfactory ability for postnatal development. Our data show that OSNs expressing oORs are likely to mature and be functional in RTP1,2DKO, explaining the residual responses observed in the mutant. These OSNs become more abundant in the mutant OE as the mouse ages indicating that these ORs help maintain the olfactory ability of these mice. It will be interesting to assess olfactory-mediated behaviors of the RTP1,2DKO mice in which only a minor fraction of the ORs are functional, since this could address a fundamental question of why most mammals have so many ORs.

No GFP-positive glomeruli were observed in RTP1,2DKO;M71-IRES-tauGFP indicating OSNs expressing Olfr151 are unable to converge their axons without RTP1 and RTP2. Yet we observed small tdTomato-positive glomeruli in RTP1,2DKO; M71-IRES-cre; Rosa26-lox-stop-lox-tdTomato. These tdTomato-positive glomeruli could be formed by OSNs that initially chose Olfr151 and then switched and/or stabilized the expression of the same OR, presumably an oOR. Previous reports have shown that OR gene switching tend to take place within the same gene locus (*Roppolo et al., 2007*; *Pacifico et al., 2012*) leading us to test the hypothesis that the tdTomato-positive axons forming these glomeruli in RTP1,2DKO mice could be stabilizing the expression of Olfr143, an oOR near the Olfr151 gene locus. However, we found no tdTomato-positive OSNs that also expressed Olfr143. This suggests that at least a portion of tdTomato positive OSNs stabilize the expression of the same OR, which is probably an oOR other than Olfr143. Further investigation of the identities of tdTomato-positive OSNs could further our understanding of the OR gene switching mechanism.

## Functional ORs expressed outside olfactory system

A number of reports have shown that ORs are expressed in various organs outside the olfactory system (*Griffin et al., 2009*; *Feldmesser et al., 2006*; *Flegel et al., 2013*; *Kang and Koo, 2012*), whereas expression of RTP1 and RTP2 appears to be confined to the peripheral olfactory tissues. One well-established example of a functional OR outside the olfactory system is Olfr78 (also known as MOR18-2) and its human ortholog OR51E2, which have been reported to function in the prostate, airway and kidney as well as the carotid body (*Aisenberg et al., 2016*; *Chang et al., 2015*; *Wang et al., 2006*; *Pluznick et al., 2013*). This receptor is an oOR and is trafficked to the cell surface in heterologous cells without the RTPs (*Zhou et al., 2016*; *Pluznick et al., 2011*; *Neuhaus et al., 2009*). It is also interesting to note that ORs expressed in the bladder and thyroid (Olfr544, Olfr558, and Olfr1386) are all oORs (*Kang et al., 2015*). Together, it is tempting to speculate that oORs expressed outside the OE are more likely to play chemosensory roles (*Feingold et al., 1999*).

In conclusion, our study suggests the importance of OR trafficking by the RTP family members in OSN function and the probability of OR gene choice. Our studies contribute to a broader understanding of how cells discern the presence of a GPCR on their cell surface post translationally and link protein trafficking to epigenetic modifications that give rise to changes in the cell's expression profile. In the future, it will be interesting to ask whether the UPR pathway that links the functional trafficking of receptors to the cell surface with epigenetic modifications is utilized by other tissue types.

# Materials and methods

## Generation of RTP1 and RTP2 double knockout mice

The strategies for generating RTP1 and RTP2 knockout mice are illustrated in *Figure 1*. The coding regions of RTP1 and RTP2 were replaced by loxP and puromycin (Pac) cassette, respectively. Fragments used for the left and right arms were amplified by PCR using BAC clone derived from C57BL/6 mice as templates. For RTP1, ACN cassette and DT-A cassette were used for positive and negative selection, respectively. A targeting vector was electroporated into the ES cell line EF-1, which is 129/B6 hybrid (*Bronson et al., 1996*). Colonies were picked up in G418-containing medium. Genomic DNA was digested by Acc65 I and hybridized with a 500 nt external probe on Southern blots. To generate RTP1&2 double knockouts, the RTP2 targeting vector was electroporated into two ES cell lines in which RTP1 is replaced by loxP-Cre-neo-loxP cassette. Colonies were picked up in puromycin-containing medium. Genomic DNA was digested by Bgl II and hybridized with a 500 nt external probe on Southern blots. Targeted ES cell clones were injected into C57BL/6 blastocysts. Chimeric mice were bred with C57BL/6 mice. Mice with RTP1,2 mutant allele were maintained by backcrossing with C57BL/6. Primers for genotyping are: 5'-cggaattcatgtcaggctgcaacttc-3' and 5'-gggccga-tattgggttaggag-3' for WT allele; 5'-agccagctcttaagtccttac-3' and 5'-gctcgagatctagatatcgataccgt-3' for mutated allele. Primers for genotyping RTP2 KO are: 5'-ccctgaagagtctcacccgctc-3' and 5'-cacata-taccccaacttctagg-3' for WT allele; 5'-caaacagacgaaccctagcaattcccactg-3' and 5'-cttcattctcag-tattgttttgccaagttc-3' for mutated allele.

## Animal strains

The procedures of animal handling and tissue harvesting were approved by the Institutional Animal Care and Use Committee of Duke University. All experiments were carried out on both male and female mice. The following mouse strains were all obtained from The Jackson Laboratory: Olfr151[tm14(Adrb2)Mom]/MomJ ($\beta_2$ Adrenergic Receptor-IRES-LacZ) (*Feinstein et al., 2004b*), Stock no. 006691 (RRID:IMSR_JAX:006691). B6;129P2-Olfr151[tm26Mom]/MomJ (M71-IRES-tauGFP) (*Feinstein et al., 2004a*), Stock no. 006676 (RRID:IMSR_JAX:006676). Olfr151[tm28(cre)Mom]/MomJ (M71-IRES-Cre) (*Li et al., 2004*), Stock no. 006677 (RRID:IMSR_JAX:006677). B6;129S6-Gt(ROSA) 26[Sortm9(CAG-tdTomato)Hze]/J (ROSA26-loxP-stop-loxP-tdTomato) (*Madisen et al., 2010*), Stock no. 007905 (RRID:IMSR_JAX:007905).

## DNA constructs

OR ORFs were cloned into pCI (Promega) with a Rho tag at the N terminal. All plasmids were verified using Sanger's sequencing (3100 Genetic Analyzer, Applied Biosystems).

## Electroolfactrograms (EOG)

Mice aged 2–4 months were sacrificed by anesthesia followed by rapid decapitation. After removing the skin, the skull was hemisected along the midline with a razor blade, exposing the turbinates. Hemisections were stabilized with pins in a custom Sylgard chamber for recording. Electroolfacto-grams were measured using glass pipettes filled with ACSF (tip size15–20μm, resistance ~0.5 MΩ). For some recordings, electrode tips were filled with 0.5% agarose. Electrodes were placed on the anterior surface of turbinate II and referenced to an Ag/AgCl wire placed on the surrounding bone. Signals were filtered (0.1Hz - 100 Hz) and amplified (1000X) using a differential amplifier (DAM-80, WPI). High-pass filtering introduced a slight rebound above baseline for strong responses. Odors were delivered using a custom olfactometer at a final dilution of 0.01% (1:1000 vol/vol in mineral oil, and a further 1:10 via airflow by combining flow from odorant headspace with a moisturized deodor-ized airstream) and a total flow rate of 100 ml/min. Odorants were obtained from Sigma at the high-est purity available and diluted in mineral oil. Epithelia were kept moisturized at all times. The blanks are an average of multiple interleaved trials interspersed within the series. This averaged blank is re-displayed for each different odorant for comparison.

## Immunofluorescence

Mice were weaned at three weeks and their OE was dissected out and flash frozen. 20–25 micron thick sections were cut onto slides which were stored at −80°C and were subsequently thawed and fixed in 4% PFA for 20 min, washed for 1 min with 0.5% triton-x-100 in PBS and then rinsed twice in PBS. They were blocked for 1 hr (See table for blocking reagent and antibody concentration) in PBS with 0.1% triton-x-100 and then kept overnight at 4°C in primary antibody solution made in the same blocking reagent. The antibody solution was washed 3 times for 5 min in PBS and then subject to the secondary antibody for an hour. 0.001% bisbenzimide, used to visualize the nucleus, was added to the slides for 1 min followed by 3 × 5 min washes in PBS. The coverslip was mounted in 5–6 drops of Mowiol.

| No. | Target | Source | Dilution | Company | Blocking | Catalog no. | RRID |
|-----|--------|--------|----------|---------|----------|-------------|------|
| 1 | GFP | Rabbit | 1:400 | | 5% milk | | |
| 2 | OMP | Goat | 1:400 | Wako | 5% milk | 019-22291 | AB_664696 |
| 3 | pS6 | Rabbit | 1:200 | ThermoFisher | 5% milk | 44-923G | AB_2533798 |
| 4 | Caspase 3 | Rabbit | 1:1000 | Cell Signaling | 5% milk | 9661 | AB_2341188 |
| 5 | LSD1 | Rabbit | 1:800 | abcam | 4% Donkey serum | ab17721 | AB_443964 |
| 6 | ATF5 | Goat | 1:1000 | Santa Cruz | 4% Donkey serum | sc-46934 | AB_2058761 |
| 7 | M71 | Guinea Pig | 1:3000 | Barnea et al. | 4% Donkey serum | | |
| 8 | LacZ | Mouse | 1:3000 | Promega | 4% Donkey serum | Z3781 | AB_430877 |

| 9 | Td Tomato | Rabbit | 1:10,000 | Rockland | 4% Donkey serum | 600-401-379 | AB_2209751 |

## RNA in situ hybridization

Candidate ORs which had less than 80% identity with all other ORs were chosen from the RNA-Seq data. Probes against their ORFs were prepared by the addition of the T3 start site to the 3' end of the pCI plasmid primer followed by incorporation of digoxigenin (DIG) using T3 RNA polymerase (Promega) and alkaline degradation to get labeled probes of around 200 bp. Slides were prepared as described above and fixed in 4% PFA for 15 min and washed twice in PBS. They were acetylated in 1.2% triethanolamine and dropwise addition of acetic acid. They were washed in PBS for 5 min and prehybridized in buffer for an hour in large plates soaked in 5XSSC and 50% formamide at 58°C. 1 μl of the aforementioned labeled probe in 200 μl of the prehybridization buffer was pipetted on to the slides and covered with a layer of parafilm and kept overnight at 58°C. The slides were thoroughly washed in 5XSSC and then in 0.2XSSC twice for 30 min, 5 min in PBS and then blocked for an hour. The slides were stained with alkaline phosphatase conjugated antibody against DIG (Roche) for an hour and then kept in development buffer for 5 min before being subject to NBT-BCIP in development buffer overnight. Slides were then stained with bisbenzimide and mounted in Mowiol as described above.

## Fluorescent in situ hybridization (FISH)

Slides were prepared and treated using the in situ hybridization protocol until the antibody staining step. Horse radish peroxidase (HRP) conjugated antibody against DIG was used instead of the alkaline phosphatase conjugate. Antibody staining amplification was carried out using tyramide signal amplification (TSA) using cy3 as the flourophore (PerkinElmer) followed by antibody staining against nATF5 as described in the immunofluorescence section above.

## Whole mount LacZ staining

Three-week-old mice were sacrificed and the entire head was dissected and kept in 4%PFA for 30 min. The tissue was washed in buffer A (100 mM phosphate buffer [pH 7.4], 2 mM MgCl2, and 5 mM EGTA) once for 30 min and then for 5 min in buffer A and buffer B (100 mM phosphate buffer [pH7.4], 2 mM MgCl2, 0.01% sodium desoxycholate, and 0.02% Nonidet P40) and then kept in buffer C (buffer B, with 5 mM potassium-ferricyanide, 5 mM potassium-ferrocyanide, and 1 mg/ml of X-Gal) containing X gal overnight at 4°C. Whole mount tissue was then washed in PBS and imaged under a 5x objective (*Mombaerts et al., 1996*). The medial glomeruli were not observed using this method.

## RNA-Seq

The whole olfactory mucosa was dissected out of 3-week-old mice (2 males and 1 female sex matched littermates) and homogenized in trizol. This solution was centrifuged at maximum speed for 10 min and the supernatant collected was treated with 0.2 ml chloroform for every 1 ml of trizol, hand shaken for 3 min and spun for 15 min at maximum speed. The aqueous phase was collected and added to isopropanol (500 μl per 1 ml of trizol) and shaken and kept for 5 min at room temperature and then spun for 10 min. The RNA pellet thus collected was washed once in 75% ethanol by adding 500 μl per 1 ml of trizol and centrifuged for 2 min and then again in 180 μl of 75% ethanol. The pellet was then briefly air-dried before being suspended in 50 μl of water and the concentration of RNA was determined using a spectrophotometer by taking 1 μl of the sample and diluting it with 9 μl of water. RNeasy cleanup kit was then used to process the sample. Library generation was carried out using Illumina TruSeq Stranded RNA-Seq kit and HiSeq Illumina sequencing was carried out at the Duke Sequencing and Genomic Technologies Core.

Reads were mapped, using kallisto (*Bray et al., 2016*), to the mouse transcripts which were downloaded from UCSC Genome Browser (https://genome.ucsc.edu) and whose OR genes were replaced with extended OR gene annotations. Reads assigned on each gene were counted by kallisto. The read count table thus generated was analyzed using EdgeR. DESeq was used to calculate the size factors of individual libraries, FDR (False Discovery Rate) was used to adjust multiple comparisons between the ORs as previously published (*Jiang et al., 2015*).

## Cell culture

HEK293T cells were grown in Minimal Essential Medium (MEM) containing 10% FBS (vol/vol) with penicillin-streptomycin and amphotericin B at 37°C, saturating humidity and 5% $CO_2$. These cells were authenticated using polymorphic short tandem repeat (STR) at the Duke DNA Analysis Facility using GenePrint 10 (Promega) and shown to share profiles with the reference (ATCC). No mycoplasma infection was detected.

## FACS

HEK293T cells were grown to a 100% confluence before being trypsinized and resuspended and seeded onto 35 mm plates at 25% confluency. These plates were cultured overnight then transfected with rho tagged ORs in the plasmid pCI for 18 to 20 hr along with GFP expression vector to monitor the transfection efficiency. The cells were stripped with cell stripper and triturated before being kept in 5 mL round bottom polystyrene (PS) tubes (Falcon 2052) on ice. The cells were spun down at 4°C and resuspended in PBS containing 10 mM HEPES, 15 mM NaN3, and 2% FBS to wash the cell stripper. They were subjected to 30 min in primary antibody (mouse anti Rho [*Laird and Molday, 1988*]) and then washed, stained with phycoerythrin (PE)-conjugated donkey anti-mouse antibody (Jackson Immunologicals) in the dark. 7-Amino-actinomycin D (Calbiochem), a fluorescent, cell-impermeant DNA binding agent that selectively stains dead cells, was added to eliminate dead cells. The cells were analyzed using BD FACSCanto II FACS with gating allowing for 10,000 GFP positive, single, spherical, viable cells and the results were analyzed using Flowjo (*Dey and Matsunami, 2011*).

## Image analysis

All images were captured on Zeiss Axioskop two fluorescent microscope using Q capture pro. The images were then analyzed using ImageJ. For OR bias, phosphorylated S6 and caspase three experiments, nuclear staining was quantified by selecting the OSN layer in the OE and using the maxima function in ImageJ to select and count all cells followed by quantification of OR or Caspase three positive OSNs by hand scoring using the cell counter function. Percent positive cells were calculated as (Positive cells/ total number of cells)*100. ATF5 was quantified by selecting the OSN layer and digitally straightening it using ImageJ followed by manually selecting the ATF5 positive cells and using the measure function to get the X and Y co-ordinates to measure the height from the basal end of the epithelium. All colocalization experiments were manually scored by selecting the OR positive cell's nucleus and measuring the pixel intensity for the same selection. The pixel intensity of the neighboring area was subtracted to remove background and determine positive cells.

## Marker gene analysis

The area occupied by OSNs in the OE was hand selected using nuclear staining in image J. The thickness of the sections was measured by straightening the OE followed by measuring the height of the straightened section in four places. The average height for every position was compared between the two genotypes. The OMP or ACIII or GAP43 positive area was selected using imageJ thresholding and these areas were measured and expressed as percentages in the figures. Images were taken at five roughly equivalent positions in the OE using the VNO and OB as landmarks (anterior VNO, middle VNO, posterior VNO, anterior OB and middle OB, *Figure 2—figure supplement 1* shows an example of matching OE sections using the anterior OB). We further analyzed all the sections from RTP1,2DKO mice at 1 day, 21 day and 6 months and compared the difference in percent area occupied by the OMP positive layer, ACIII positive layer and GAP43 positive layer with that of their littermates. Each data point is obtained from one image from one matched section. The data sets were compared using a paired student *t* test or Mann Whitney *U* test as indicated in the figure legend.

## Statistical analysis

Percent positive cells were calculated by hand scoring positive cells and calculating percentage based on the total number of OSNs in the image counted using nuclear staining. Each individual section was counted as an individual data point. Height of positive cells was calculated by straightening

OE sections and obtaining the Y co-ordinates of hand scored ATF5 positive cells. Multiple comparison data from our ANOVA analysis is included in *supplementary file 3*.

### pS6 staining quantification
Each OSN positive for OR FISH signal was selected in image J. These selections were used to measure the pixel intensity of pS6 staining. The average pixel intensity of the entire OE selection was used as background. The average intensity of each cell was normalized by the background followed by its subtraction.

## Acknowledgements

We would like to thank Gilad Barnea and Richard Axel for anti-M71 antibodies. Cheryl Bock and other members of Duke Cancer Institute Transgenic Mouse Facility for ES cell targeting and chimeric mouse production; Mengjue Jessica Ni for expert technical assistance. Simone Weyand, Ting Zhou, Claire de March, Xiaoyang Serene Hu, Tatjiana Abaffy, Kevin Zhu, Aashutosh Vihani, along with other members of the Matsunami lab and for critical input and discussion of the experiments and manuscript; William Wetsel for his help with statistical analysis; Doug Marchuk and Debby Silver for generously sharing equipment; Jianghai Ho and Mike Cook in Duke Flow Cytometry Core for help with FACS analysis and Duke Sequencing and Genomic Technologies Core for carrying out the RNA-Seq. This work was supported by grants from NIH.

## Additional information

### Funding

| Funder | Grant reference number | Author |
|---|---|---|
| National Institutes of Health | R01 DC014423 | Hiroaki Matsunami |
| National Institutes of Health | R01 DC012095 | Hiroaki Matsunami |

The funders had no role in study design, data collection and interpretation, or the decision to submit the work for publication.

### Author contributions
RS, YI, Conceptualization, Data curation, Formal analysis, Investigation, Methodology, Writing—original draft, Writing—review and editing; G, Data curation, Formal analysis, Funding acquisition, Investigation, Writing—original draft, Writing—review and editing; KI, Data curation, Software, Formal analysis, Investigation, Writing—original draft, Writing—review and editing; M-SC, HY, QC, MK, Data curation, Investigation, Writing—review and editing; MY, ME, Formal analysis, Supervision, Funding acquisition, Investigation, Writing—review and editing; HM, Conceptualization, Data curation, Formal analysis, Supervision, Funding acquisition, Investigation, Visualization, Methodology, Writing—original draft, Project administration, Writing—review and editing

### Author ORCIDs
Ruchira Sharma, http://orcid.org/0000-0002-2795-7457
Ian Davison, http://orcid.org/0000-0003-0998-7676
Hiroaki Matsunami, http://orcid.org/0000-0002-8850-2608

### Ethics
Animal experimentation: This study was performed in strict accordance with the recommendations in the Guide for the Care and Use of Laboratory Animals of the National Institutes of Health. All of the animals were handled according to approved institutional animal care and use committee (IACUC) protocols (A161-16-07) of the Duke Animal Care and Use program.

## Additional files

### Supplementary files

• Supplementary file 1. The number of sequence reads that map each annotated gene in RNA-Seq from three wild-type and 3 RTP1,2DKO mice. FDR is calculated against the entire data set and the fold change (logFC) is displayed as the log (average wild-type reads/ average RTP1,2DKO reads).

• Supplementary file 2. List of oORs and uORs with their chromosomal locations and their expression zones (dorsal versus ventral) in the OE.

• Supplementary file 3. Results of one-way ANOVA and Tukey's post hoc tests

### Major datasets

The following dataset was generated:

| Author(s) | Year | Dataset title | Dataset URL | Database, license, and accessibility information |
|---|---|---|---|---|
| Matsunami H, Sharma R, Ikegami K | 2017 | Olfactory receptor accessory proteins play crucial roles in receptor gene choice | https://www.ncbi.nlm.nih.gov/geo/query/acc.cgi?acc=GSE87694 | Publicly available at the NCBI Gene Expression Omnibus (accession no: GSE87694) |

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
