## [Decision Letter]

Thank you for submitting your manuscript "Olfactory receptor accessory proteins play crucial roles in receptor function and gene choice" to *eLife*. Your article has been reviewed by three peer reviewers, and the evaluation has been overseen by Catherine Dulac as the Senior Editor and a Reviewing Editor. The following individual involved in review of your submission has agreed to reveal his identity: Ron Yu (Reviewer #3). As you will see, all of the reviewers were impressed with the importance and novelty of your work.

The three reviews (lightly edited) are included at the end of this letter, as there are a variety of specific and useful suggestions in them. The reviewers had specific technical concerns (that you can likely address) regarding the data. Importantly, they also considered the data presented to be insufficient to uniquely support your model for a direct active role for RTPs in the choice process that directs expression of one OR within each individual single OSN. In the absence of additional data that rule out alternate models (described in the reviewers' comments), the alternate models should be discussed.

*Reviewer #1:*

The majority of mammalian odorant receptors do not traffic to cell membrane in heterologous expression systems. The authors' group previously identified RTP1 and RTP2, which are specifically expressed in OSNs and promote surface expression of ORs in heterologous systems (Saito et al., Cell 2004). The authors now address the function of RTP1 and RTP2 in vivo by using RTP1,2 double knockout mice. Generation of the double knockout is not trivial as the two genes are 500 kb apart on the same chromosome. The authors achieved this by consecutive manipulation of the same ES cell. The authors show that OR proteins, at least for OR M71, accumulated in the soma and do not traffic to the dendrite and cilia in RTP1,2(-/-) OSNs, thus demonstrating that RTP1 and RTP2 are necessary for proper OR trafficking in OSNs. Consistently, RTP1,2(-/-) olfactory epithelium exhibits substantially reduced odor response, and M71 OSNs exhibit axon targeting defect. The authors also show that the RTP1,2(-/-) OE has less mature OSNs, increased cell death and altered representation of OR transcripts with about half of OR genes being downregulated and, very interestingly, 8% OR genes upregulated. The authors further conducted experiments examining the unfolded protein response, a process that has been suggested to play roles in OR gene choice, by correlating OR expression with nATF5 translation and the stability of OR gene choice by a clever lineage tracing strategy. The results of these experiments support a role of RTP1,2 in alleviating UPR and stabilizing OR gene choice for ORs that are dependent on RTP for surface trafficking.

This is a significant study addressing questions not only fundamental to the olfactory field, but also important to GPCR trafficking, cell survival and gene expression regulation in general. The experiments are well designed and well conducted in large. Although many more experiments need to be done to fully address the relationship between RTP1,2 and OR gene and protein, the current work stands as a significant progress toward this goal. I have several comments for the authors to consider, mostly concerning technical and/or interpretational issues.

1) The evidence for a role of RTP1,2 in regulating OR gene choice can be complicated by the substantial cell death phenotype in RTP1,2(-/-) mice. A biased cell death alone, independent of the change in probability of OR gene choice, could sufficiently alter OR population representation both in the level of transcripts and in the number of OSNs. Using, say, Bax knockout mice might help to address this issue, but this would be too much for the current work. Although the authors have beautifully done the OSN lineage tracing experiment and the ATF5 correlation experiment, it is still important to carefully discuss how cell death may influence (exaggerate) the observed changes in OR population representation.

2) Figure 1 should also provide the total thickness of OE and a ratio of the OMP layer vs. the total. The OE thickness varies upon location, more detailed information should be provided about these quantification analyses.

3) The largely diminished odor response is quite striking in RTP1,2(-/-) mice, despite the OE being overly populated by presumably those oORs encoded by 8% of total OR genes. Are any of the 7 odorants a ligand of those oORs? In the subsection “Odorant evoked electrophysiological responses in RTP1,2(-/-) mice are diminished” the authors state, "some sensitivity was maintained for a subset of odorants (2-heptanone, amyl acetate, isomenthone)… (Figure 2)." However, Figure 2 shows that all 7 odorants give rise to noticeable responses. How was the quantification done here? Also, what does Figure 2 tried to show? Is it the response to the odor mix? Arithmetic cumulative of individual odorants may not be appropriate.

4) In the second paragraph of the subsection “OR expression is biased in RTP1,2(-/-)”, the authors wrote, "A majority of ORs (559/1098) were underrepresented". It may be better to use the term "half of" to describe this 51% of OR population. Similarly, in the first paragraph of the subsection “Persistent expression of nATF5 is observed in RTP1,2(-/-) OSNs expressing uORs”, "with most ORs being underrepresented".

5) The result in Figure 4 is interesting, where the β2-AR is markedly underrepresented compared to M71 in RTP(+/+) mice, even though they are both from the same M71 locus. How would this result fit to the authors' model?

6) A significant portion (41%) of OR genes exhibit no change in the level of transcripts in RTP1,2(-/-) mice. This portion of ORs is completely ignored in the manuscript. It is proper to at least clearly mention this portion of ORs in the Results and discuss how this unchanged level may relate to the RTP function.

*Reviewer #2:*

The manuscript by Sharma et al. provides important in vivo evidence consistent with for a role of the RTP proteins in trafficking of ORs in mouse olfactory epithelium. For the most part, the data presented is clear and of high quality. The major concern regarding the manuscript centers on the authors interpretation of the data to infer a direct, active role of the RTP proteins in OR choice.

The authors have previously demonstrated a role for the RTP proteins in the trafficking of OR proteins in heterologous systems. This evidence, combined by the selective expression of these proteins in OSNs, suggest that a similar important role exists in vivo. Previous evidence from several laboratories, including the author, suggest that RTPs are required for, or contribute significantly to, the plasma membrane trafficking of some ORs expressed from CMV promoters in modified HEK cells while other ORs can achieve significant levels of surface expression in the absence of these proteins. Presumably, in these heterologous cell lines the process of OR choice is not operative. Given the requirement for trafficking of ORs to OSN surface membranes to achieve proper axon targeting and cell survival, the most likely explanation is that RTPs produce the observed selective enrichment of some ORs (oORs) over others (uORs) through a passive mechanism.

All of the author's data appears to be consistent with OSNs expressing ORs that are relatively independent of RTP are preferentially retained, while those that require RTP are preferentially lost. As subsequent continued proliferation of progenitors proceeds, those that stochastically select RTP dependent ORs will again have limited life span/viability. Over time, this will enrich the tissue for the observed subset of all ORs. Other data, including the findings from b-2 AR are consistent with this model. Finally, the persistent expression of nATF5 in the M71-expressing population of cells reflects the fact that these cells are immature and fated to shortly die. This model contrasts with that proposed in the Abstract and the body of conclusions of the paper – “We present a model in which developing OSNs exhibit unstable OR expression until they choose to express an OR that exits the ER. Our study sheds light on the new link between OR protein trafficking and OR transcriptional regulation”. Instead it suggests simply that cells expressing ORs that cannot traffic efficiently in the absence of RTPs die/turnover and ones that can traffic in the absence of RTPs accumulate.

*Reviewer #3:*

In this study, Sharma and colleagues examined odorant receptor (OR) expression in the mouse olfactory epithelium of RTP1 and RTP2 knockout mice. They found that mutation of both genes results in elevated apoptosis and the reduction of mature OSNs. OR trafficking to the dendrites are affected in a subpopulation of OSNs. Global survey and in situ hybridization indicate a change in the representation of the ORs expressed. The authors further found that molecules involved in regulating the monoalleleic expression of OR genes, including ATF5 and LSD1, are dysregulated. The authors conclude that RTP family of proteins affect OR transcription regulation and receptor gene choice by regulating OR protein trafficking.

The experiments were carried out in a logical fashion and the data quality is high. The evidence clearly establishes that affecting OR protein trafficking causes dysregulation of genes involved in receptor choice and instability of receptor gene expression. The data support current model of receptor choice and reveal the involvement of molecular chaperones in the process.

My main concern is conceptual. Although it is manifest as population frequency of expression in the OE, receptor choice is fundamentally a process at single cell level. For this study, the authors appear to have conflated the two. It is not clear to what extent the redistribution of OR expression reflects the change in receptor gene choice at single cell level per se. There are two potential mechanisms that can result in the underrepresentation of some ORs and overrepresentation of others at the population level. First, RTPs have direct impact on receptor gene choice at individual cell levels, by facilitating receptor protein transport and stabilizing receptor gene expression. In the absence of RTPs, OR gene expression is only stabilized for receptors that do not require RTPs for surface expression is expressed. In this scenario, the cell keeps switching ORs until an oOR is expressed. Therefore, the overrepresentation of oORs is the result of switching that settles on the oOR. Alternatively, protracted instability of receptor gene expression leads to apoptosis, resulting in the loss of cells that express ORs requiring RTP. In this scenario, the redistribution of OR expression is the result of displacement of these dying neurons by those expressing ORs not requiring RTPs. The authors clearly favor the former as they conclude as such, when it is equally possible that the second mechanism can completely or partially account for the observations.

It is likely that both mechanisms are at play. The authors should discuss these possibilities and clarify their position. Additional analyses listed below should also help achieving a better understanding of the observations:

1) Because transcriptome analyses are performed on the entire OE, the diminished number of OSNs may skew the actual change of OR expression in the OSN population. The FPKM values should be normalized against mature OSN markers such as OMP, Gnal or CNGA2, to show OR expression change in the OSNs.

2) It is widely accepted that switching of OR gene expression occurs preferentially among genes clustered within the same locus. I suggest the author to perform in situ hybridization of oORs (e.g. olfr143) in the M71-Cre marked mice. According to the author's model, it is expected that the frequency of oOR expressing will increase in the M71-Cre marked cells.

3) The analyses were all performed in 3 week old mice. No developmental changes were captured. The authors should perform in situ hybridization of a few uOR and oOR in early postnatal and adolescent mice. If cell death is the main drive of redistribution of OR expression in the RTP mutant, it would be expected that at early developmental stages the effect is less striking whereas at later stages it is more pronounced.

---

## [Author Response]

*Reviewer #1:*

*[…] This is a significant study addressing questions not only fundamental to the olfactory field, but also important to GPCR trafficking, cell survival and gene expression regulation in general. The experiments are well designed and well conducted in large. Although many more experiments need to be done to fully address the relationship between RTP1,2 and OR gene and protein, the current work stands as a significant progress toward this goal. I have several comments for the authors to consider, mostly concerning technical and/or interpretational issues.*

*1) The evidence for a role of RTP1,2 in regulating OR gene choice can be complicated by the substantial cell death phenotype in RTP1,2(-/-) mice. A biased cell death alone, independent of the change in probability of OR gene choice, could sufficiently alter OR population representation both in the level of transcripts and in the number of OSNs. Using, say, Bax knockout mice might help to address this issue, but this would be too much for the current work. Although the authors have beautifully done the OSN lineage tracing experiment and the ATF5 correlation experiment, it is still important to carefully discuss how cell death may influence (exaggerate) the observed changes in OR population representation.*

We agree with the reviewer that cell death alone could alter OR population representation both in the level of transcripts and in the number of OSNs, and additional experiments with Bax knockout would clarify this issue. We think that both biased cell death of uOR‑expressing OSNs and OR gene choice contribute to an increased number of oOR-expressing OSNs in RTP1,2 (‑/-). We agree with the reviewer in carefully discuss the role of cell death in OR representation. We have addressed these points in the Discussion as follows:

“Both change in probability of OR gene choice and biased cell death could alter OR population representation both in the level of transcripts and in the number of OSNs. The relative contributions of cell death and gene switching to the differential representation of ORs can be further clarified using Bax knockout mice where cell death in developing neurons is suppressed (Robinson, Conley and Kern, 2003).”

*2) Figure 1 should also provide the total thickness of OE and a ratio of the OMP layer vs. the total. The OE thickness varies upon location, more detailed information should be provided about these quantification analyses.*

The reviewer is correct that the OE thickness varies. Therefore, we used sections from equivalent positions of OE along the anterior-posterior axis. We have carried out new experiments to quantify the ratio of the area occupied by OMP compared with the total area occupied by OSNs. We have confirmed that RTP1,2DKO have a smaller fraction of OMP positive cells, represented by the area occupied by the RNA in situ hybridization, in different parts of the OE. We have added the following to our Results:

“Upon examination of the OE we found that its thickness was significantly reduced in RTP1,2DKO mice. (p=0.02 paired student t test) (Figure 2). […] Comparison of the OMP positive layer from wild-type and RTP1,2DKO OE collected at 1 day old, 21day old and 6 month old mice showed a significant reduction in OMP expression at 1 day and 21 days (p=0.0003, Mann Whitney U test, p=0.0003, Mann Whitney U test) but not at 6 months (Figure 2).”

*3) The largely diminished odor response is quite striking in RTP1,2(-/-) mice, despite the OE being overly populated by presumably those oORs encoded by 8% of total OR genes. Are any of the 7 odorants a ligand of those oORs? In the subsection “Odorant evoked electrophysiological responses in RTP1,2(-/-) mice are diminished” the authors state, "some sensitivity was maintained for a subset of odorants (2-heptanone, amyl acetate, isomenthone)… (Figure 2)."*

In order to address the reviewers concern about the functionality of oORs, we must identify active ligands for oORs. Since we do not have available data for active ligands for most of the ORs, we decided to take advantage of the data that we recently published in Jiang et al. (2015) in which we identified ORs activated by two odorants, acetophenone and 2,5-dihydro-2,4,5- trimethylthiazoline (TMT), in vivo. We selected a uOR, olfr923, that robustly responded to acetophenone, and an oOR, olfr1395, that robustly responded to TMT. We carried out further experiments using these ORs and obtained data consistent with the idea that OSNs expressing oORs but not uORs respond to their cognate ligands in RTP1,2(‑/-) mice. We have detailed our findings as follows:

“To evaluate function of OSNs expressing uORs or oORs, we chose a uOR and an oORs that have been previously deorphanized. Olfr1395 is an oOR found to respond to 2,5-dihydro-2,4,5- trimethylthiazoline (TMT) and Olfr923, a uOR, to acetophenone in vivo (Jiang et al., 2015). […] In contrast OSNs expressing Olfr923 in het, but not RTP1,2DKO were activated by their cognate ligand acetophenone (Figure 7). These data show that OSNs expressing oORs mature and function in the RTP1,2DKO.”

*However, Figure 2 shows that all 7 odorants give rise to noticeable responses. How was the quantification done here? Also, what does Figure 2 tried to show? Is it the response to the odor mix? Arithmetic cumulative of individual odorants may not be appropriate.*

Quantification of the odorant response was carried out by subtracting the peak blank response from the peak response obtained by the odorant. We revised both our Methods section (subsection “Electroolfactrograms (EOG)”), Results (subsection “Odorant evoked electrophysiological responses in RTP1,2DKO mice are diminished”) and the Figure 3 legend 3. We agree that Figure 2 is confusing and have therefore removed it from our manuscript.

“The blanks are an average of multiple interleaved trials interspersed within the series. This averaged blank is re-displayed for each different odorant for comparison.”

“Responses to most odors were identical to the blank stimulus (air only), although some sensitivity was maintained for a subset of odorants (2-heptanone, amyl acetate, isomenthone) compared to the wild-types (Figure 3).”

“(C) Quantification of the EOG amplitudes for each of the 7 odorants showing that only a few of the odors elicit responses from the RTP1,2DKO OE and these responses are lower than the wild-type. Each bar represents the difference between the peak of the odor minus the peak of the air only blank.”

*4) In the second paragraph of the subsection “OR expression is biased in RTP1,2(-/-)”, the authors wrote, "A majority of ORs (559/1098) were underrepresented". It may be better to use the term "half of" to describe this 51% of OR population. Similarly, in the first paragraph of the subsection “Persistent expression of nATF5 is observed in RTP1,2(-/-) OSNs expressing uORs”, "with most ORs being underrepresented".*

Based on new sequence mapping, we have obtained updated estimates which show that 531/1088 ORs are underrepresented constituting 49%. We have changed our Results and Discussion sections to read “close to half of”.

“Close to half of the annotated intact ORs (531/1088) were downregulated in RTP1,2DKO (FDR corrected p<0.05), consistent with fewer OSNs in the mutant.”

“The absence of RTP1 and RTP2 leads to the underrepresentation of nearly half of the ORs, while about 10% of the ORs are significantly overrepresented.”

*5) The result in Figure 4 is interesting, where the β2-AR is markedly underrepresented compared to M71 in RTP(+/+) mice, even though they are both from the same M71 locus. How would this result fit to the authors' model?*

I agree with the reviewer that this is an interesting observation, because it suggests that the amino acid sequence of an OR influences the initial frequency with which it is chosen leading to different ORs being differentially abundant in the OE. That said, we would like to interpret the data with caution, as β2-AR is not an OR. We have addressed the reviewer’s comment

in the Discussion as follows:

“RNA-Seq data in the wild-type suggests that oORs as a group tend to be more frequently expressed. It could be that initial expression of RTP1 and RTP2 in the developing OSNs is not stable or abundant enough, causing oORs to be stabilized. […] The relative contributions of cell death and gene switching to the differential representation of ORs can be further clarified using Bax knockout mice where cell death in developing neurons is suppressed (Robinson, Conley and Kern, 2003).”

*6) A significant portion (41%) of OR genes exhibit no change in the level of transcripts in RTP1,2(-/-) mice. This portion of ORs is completely ignored in the manuscript. It is proper to at least clearly mention this portion of ORs in the Results and discuss how this unchanged level may relate to the RTP function.*

We appreciate the reviewers concern but we do not believe that 41% of OR genes exhibit no change in their transcript levels in RTP1,2(-/-), instead, we believe that a large portion of these ORs do not show statistical significance due to their low-level expression. In fact, log fold change of “not significant” ORs ranges from -3 to 4 (See Figure 4). To clarify our position, we added the following point to our Results and Discussion:

“Curiously, we found that oORs as a group are more abundantly expressed than uORs in the wild-type. The OR genes that were not classified as either underrepresented nor overrepresented (NS, not significant) exhibited wide range of changes in expression levels between the wild-type and RTP1,2DKO, but are expressed at significantly lower abundance levels than both oORs and uORs (Figure 4) (p<0.0001 one-way ANOVA, Tukey’s post hoc test).”

“OR genes that did not significantly change in RTP1,2DKO mice showed lower abundance in wild-type mice suggesting that these ORs are chosen less frequently. Deeper sequencing and/or increased samples sizes will help classify these ORs as underrepresented, overrepresented, or not changed.”

*Reviewer #2:*

*The manuscript by Sharma et al. provides important in vivo evidence consistent with for a role of the RTP proteins in trafficking of ORs in mouse olfactory epithelium. For the most part, the data presented is clear and of high quality. The major concern regarding the manuscript centers on the authors interpretation of the data to infer a direct, active role of the RTP proteins in OR choice.*

*The authors have previously demonstrated a role for the RTP proteins in the trafficking of OR proteins in heterologous systems. This evidence, combined by the selective expression of these proteins in OSNs, suggest that a similar important role exists in vivo. Previous evidence from several laboratories, including the author, suggest that RTPs are required for, or contribute significantly to, the plasma membrane trafficking of some ORs expressed from CMV promoters in modified HEK cells while other ORs can achieve significant levels of surface expression in the absence of these proteins. Presumably, in these heterologous cell lines the process of OR choice is not operative. Given the requirement for trafficking of ORs to OSN surface membranes to achieve proper axon targeting and cell survival, the most likely explanation is that RTPs produce the observed selective enrichment of some ORs (oORs) over others (uORs) through a passive mechanism.*

*All of the author's data appears to be consistent with OSNs expressing ORs that are relatively independent of RTP are preferentially retained, while those that require RTP are preferentially lost. As subsequent continued proliferation of progenitors proceeds, those that stochastically select RTP dependent ORs will again have limited life span/viability. Over time, this will enrich the tissue for the observed subset of all ORs. Other data, including the findings from b-2 AR are consistent with this model. Finally, the persistent expression of nATF5 in the M71-expressing population of cells reflects the fact that these cells are immature and fated to shortly die. This model contrasts with that proposed in the Abstract and the body of conclusions of the paper “We present a model in which developing OSNs exhibit unstable OR expression until they choose to express an OR that exits the ER. Our study sheds light on the new link between OR protein trafficking and OR transcriptional regulation”. Instead it suggests simply that cells expressing ORs that cannot traffic efficiently in the absence of RTPs die/turnover and ones that can traffic in the absence of RTPs accumulate.*

Following the reviewer’s suggestions, we have revised the Abstract and the body. We have added the following text to clarify our position:

“We present a model in which developing OSNs exhibit unstable OR expression until they choose to express an OR that exits the ER or undergo cell death.”

“In our model, the RTPs suppress UPR response by allowing ORs to exit the ER and be transported to the plasma membrane. […] Although we cannot rule out that RTPs themselves play a role in the elimination of the UPR response, the lack of increase in nATF5 observed in OSNs expressing oORs in RTP1,2DKO suggests interaction between the RTPs and the UPR pathway through ORs.”

“RNA-Seq data in the wild-type suggests that oORs as a group tend to be more frequently expressed. It could be that initial expression of RTP1 and RTP2 in the developing OSNs is not stable or abundant enough, causing oORs to be stabilized. […] The relative contributions of cell death and gene switching to the differential representation of ORs can be further clarified using Bax knockout mice where cell death in developing neurons is suppressed (Robinson, Conley and Kern, 2003).”

*Reviewer #3:*

*[…] My main concern is conceptual. Although it is manifest as population frequency of expression in the OE, receptor choice is fundamentally a process at single cell level. For this study, the authors appear to have conflated the two. It is not clear to what extent the redistribution of OR expression reflects the change in receptor gene choice at single cell level per se. There are two potential mechanisms that can result in the underrepresentation of some ORs and overrepresentation of others at the population level. First, RTPs have direct impact on receptor gene choice at individual cell levels, by facilitating receptor protein transport and stabilizing receptor gene expression. In the absence of RTPs, OR gene expression is only stabilized for receptors that do not require RTPs for surface expression is expressed. In this scenario, the cell keeps switching ORs until an oOR is expressed. Therefore, the overrepresentation of oORs is the result of switching that settles on the oOR. Alternatively, protracted instability of receptor gene expression leads to apoptosis, resulting in the loss of cells that express ORs requiring RTP. In this scenario, the redistribution of OR expression is the result of displacement of these dying neurons by those expressing ORs not requiring RTPs. The authors clearly favor the former as they conclude as such, when it is equally possible that the second mechanism can completely or partially account for the observations.*

*It is likely that both mechanisms are at play. The authors should discuss these possibilities and clarify their position. Additional analyses listed below should also help achieving a better understanding of the observations:*

We agree with the reviewer and we have clarified our position regarding this point in the Discussion:

“RNA-Seq data in the wild-type suggests that oORs as a group tend to be more frequently expressed. It could be that initial expression of RTP1 and RTP2 in the developing OSNs is not stable or abundant enough, causing oORs to be stabilized. […] The relative contributions of cell death and gene switching to the differential representation of ORs can be further clarified using Bax knockout mice where cell death in developing neurons is suppressed (Robinson, Conley and Kern, 2003).”

*1) Because transcriptome analyses are performed on the entire OE, the diminished number of OSNs may skew the actual change of OR expression in the OSN population. The FPKM values should be normalized against mature OSN markers such as OMP, Gnal or CNGA2, to show OR expression change in the OSNs.*

We agree with the reviewer that transcripts from non OSN cells may introduce confounding factors. We have therefore reanalyzed our RNA seq data set using only OR data to normalize the representation of each OR. We find that the number of uORs falls from 531 to 504, the number of oORs increase from 114 to 174 and the number of non-significant ORs falls from 444 to 411. We have added these data to the Results:

“The disparity in the abundance of transcripts for these ORs raised the possibility of a difference in probabilities of OSNs expressing each OR in RTP1,2DKO. To remove any possible confounding variables from non-OSN cells, we normalized our read counts using only reads mapped on intact ORs and found that 503/1088 were underrepresented and 175/1088 were overrepresented (FDR corrected p<0.05) (Figure 4, Figure 4—figure supplement 1).”

*2) It is widely accepted that switching of OR gene expression occurs preferentially among genes clustered within the same locus. I suggest the author to perform in situ hybridization of oORs (e.g. olfr143) in the M71-Cre marked mice. According to the author's model, it is expected that the frequency of oOR expressing will increase in the M71-Cre marked cells.*

We carried out a gene switching experiment with Olfr143. We were unable to find any neurons double positive for both tdTomato and olfr143. We have added the following to our Results:

“To determine whether the Olfr151 gene switched to an oOR within the same locus we carried out a co-localization analysis between Olfr143, an oOR within the Olfr151 locus and tdTomato under the control of M71-cre (Figure 9—figure supplement 1). We found no Olfr143 and tdTomato double positive OSNs in both RTP1,2DKO or their wild-type littermates, indicating that there is no higher likelihood of the gene switching between these ORs.”

*3) The analyses were all performed in 3 week old mice. No developmental changes were captured. The authors should perform in situ hybridization of a few uOR and oOR in early postnatal and adolescent mice. If cell death is the main drive of redistribution of OR expression in the RTP mutant, it would be expected that at early developmental stages the effect is less striking whereas at later stages it is more pronounced.*

We carried out our analysis on 1 day old, 21 day old as well as 6 month old mice. We found that the fraction of cells expressing oORs rises dramatically in 6 month old mice compared to 21 day old. We do not observe a rise in the number of OSNs expressing uORs in RTP1,2(-/‑) mice.

We observe an increase in the number of wild type OSNs expressing the uORs we tested with age. We believe this is due to the increase in the proportion of mature OSNs in older OEs. This is supported by increasing percentage of the OMP layer with increasing age (Figure 2). We have added our results to the manuscript:

“We wondered what happens to the proportion of OSNs expressing uORs and oORs in RTP1,2DKO mice at different ages. […] However, RTP1,2DKO showed no obvious increase in the fraction of cells expressing these ORs with age, while on the other hand, in RTP1,2DKO the number of neurons expressing oORs showed a dramatic increase both from 1 day old to 21 days and from 21 days to 6 months (p<0.0001 one-way ANOVA, Tukey’s post hoc test) (Figure 5) demonstrating that the RTP1,2DKO OE is progressively populated by oORs.”